# Diverse Image Priors
# for Black-box Data-free Knowledge Distillation

## Abstract

Knowledge distillation (KD) is a well-known technique for effectively transferring knowledge from an expert network (teacher) to a smaller network (student) with little sacrifice in performance. However, most KD methods require extensive access to the teacher or even its original training set, which are unachievable due to intellectual property or security concerns. These challenges have inspired *black-box data-free KD*, in which only the teacher's top-1 predictions and no real data are available. As a result, recent approaches leverage synthetic data as a replacement to real data to perform KD. However, they largely overlook a crucial factor for KD: *data diversity*, i.e., how broadly synthetic samples distribute. To address this problem, we propose Diverse Image Priors Knowledge Distillation (DIP-KD). We first synthesize *image priors* — semantically diverse synthetic images, then further optimize them to a diversity objective via contrastive learning, and finally extract soft knowledge to distill the student. We achieve state-of-the-art KD performance for the black-box data-free settings on eight image benchmarks. This is backed by our deep analysis, showing that data diversity is effectively improved, and how it facilitates KD performance. We publish the source code at https://osf.io/5mry8/?view_only=dee9e8fbcd114c34b45aa958a3aa32fa.

## 1 Introduction

An inevitable side effect of advancing artificial intelligence (AI) that often gets overlooked is the sheer size of AI models. This obstacle limits their deployability in constrained environments, such as mobile or edge devices. Fortunately, research on model compression addresses this problem through various techniques such as model pruning, quantization, or low-rank factorization. Among these techniques, *knowledge distillation* (KD), proposed by Hinton et al. (2015), is well known for its efficiency and intuitive approach.

The intuition of KD closely resembles human learning: the knowledge is transferred from an expert network (*teacher*) to a novice network (*student*). In traditional machine learning, the student learns to predict the ground-truth labels from a dataset, much like a human learner does self-studying given a collection of exercises and solutions. KD advantageously supplements this learning process with the teacher's knowledge, which is more informative than the ground-truth labels. Thus, the student can reach comparable performance to the teacher, even when it may have a lesser capacity.

However, KD methods often require the teacher's original training data for optimal knowledge transfer. Vision datasets often exhibit broad semantic variation, complex environments, and substantial intra-class diversity. This holds true even in privacy-sensitive domains or medical domains, due to the subject's condition, environments, or equipments. Nonetheless, such source datasets are often protected by stringent regulations as they are tied to security, privacy, and proprietary concerns. In fact, personal and health data, or data as intellectual properties all qualify for protection under the Database Directive (Directive 96/9/EC) and General Data Protection Regulation (GDPR) by European Parliament and Council (1996; 2016) in the EU, and under the Health Insurance Portability and Accountability Act (HIPAA) and the Data Security and Transparency Act (DSTA) by United States Congress (1996; 2016) in the US. These regulations effectively inhibit the use of traditional KD methods and pose the more challenging *data-free KD* problem.

Also crucial to KD is accessibility to the teacher model, i.e., what interfaces it exposes. Model's accessibility vary from *white-box* — accessing the intermediate features, parameters, activations, or gradients (Romero et al., 2015; Yim et al., 2017; Chen et al., 2019) to *black-box* — only the final logits or predictive probabilities (Wang et al., 2020; Nguyen et al., 2022; Vo et al., 2024). Unfortunately, the teacher's accessibility is often restricted. For instance, proprietary vision models, e.g., Vision AI (Google Cloud, 2025), Rekognition (Amazon Web Services, 2025) and Azure Vision (Microsoft Azure, 2025), return final object detection features instead of intermediate representations. Similarly, certain evaluation servers, e.g., ImageNet by Russakovsky et al. (2015), only return the top-1 prediction, not class probabilities. Retrieving deeper insights via these interfaces is usually costly or even impossible. In conjunction, while *black-box data-free KD* (BBDFKD) is a realistic problem, its contraints invalidates most KD methods and calls for a fresh approach.

Without training data, previous methods utilize synthetic images to tackle BBDFKD. In ZSDB3 (Wang, 2021), synthetic images are sampled from random uniform noise and then updated as trainable parameters. However, they have a suboptimal initialization strategy, making some classes unforgivingly unobtainable. Differently, IDEAL (Zhang et al., 2022) and DFHL-RS (Yuan et al., 2024) leverage a generator network for synthetic images. Both attempted to generate class-balanced and realistic images, but were at risk of mode collapse caused by generator overfitting, as pointed out in Yuan et al. (2024). This lack of diversity in synthetic images is detrimental to KD performance, as it limits the knowledge domain the student can learn from. In addition, the majority of these methods are restricted to using a single index instead of probabilities as label during KD, limiting the distillation signals.

**Our method.** We propose *Diverse Image Priors Knowledge Distillation* (**DIP-KD**) to tackle the BBDFKD problem, i.e., distilling the student from a black-box teacher which only returns top-1 prediction and without access to any real data. We focus on *diversity*, a pivotal factor to perform effective KD but was overlooked in previous methods. For a rigorous definition, we refer to diversity as "*how broadly the samples distribute in the data space*". We induce diversity into our KD framework through three phases: *Synthesis*: to craft a pipeline of synthetic images with diverse patterns and semantics, *Contrast*: to optimize them to be even more distinct through contrastive learning, and *Distillation*: to train a high-performance student from these images with soft distillation.

**Contribution.** In summary, we highlight our contributions as follows:

1. We propose *image priors*, synthetic images crafted with diversity in hierarchical structures, nonlinearity, and meaningful semantics.

2. We investigate contrastive learning as a fitting strategy to improve image diversity, where synthetic images are optimized to be distinguishable to each other.

3. We introduce a primer student to extract soft-probability as informative knowledge for KD, which is absent in previous BBDFKD methods.

4. Our analysis demonstrates our image priors are diverse in terms of both visuals and semantics, and show that they significantly contribute to our state-of-the-art KD performance.

## 2 Related Works

### 2.1 Knowledge Distillation

The pioneering idea in knowledge transferring from one neural network to another, intuitively, was to train the student to mimic the teacher's outputs (Bucilua et al., 2006). This idea was popularized in Hinton et al. (2015) as *knowledge distillation*, introducing the concept of *'soft'* probabilistic outputs, which are more informative than the *'hard'* class labels. Soft probabilities provide the student with *inter-class* relationships as hints for hidden knowledge, improving its performance. This foundational work has inspired a populated line of KD research: Romero et al. (2015); Chen et al. (2017); Park et al. (2019); Meng et al. (2019); Nguyen et al. (2021). Nonetheless, these methods are not effective for practical scenarios when data or model accessibility is limited.

## 2.2 Data-free Knowledge Distillation

The data availability hardship has inspired data-efficient KD approaches. While few-shot KD methods such as Wang et al. (2020); Nguyen et al. (2022); Vo et al. (2024) can reduce data requirements to a minimum, a small portion of the source dataset is still required. In practice, private datasets can be entirely closed, which will hinder these few-shot approaches. Some other approaches opt for using *proxy datasets* in the hope they are exchangeable with the unknown source dataset (Chen et al., 2021; Tang et al., 2023; Nguyen et al., 2023). However, they often have a strong built-in bias that could mismatch with the teacher trained on a different dataset, as noted in Torralba & Efros (2011). When data access is prohibited, data-free KD approaches often rely on dissecting the teacher's internal structure or leveraging its gradients as guidance to harvest knowledge (Chen et al., 2019; Fang et al., 2019; 2021; Do et al., 2022; Tran et al., 2024).

## 2.3 Black-box Data-free Knowledge Distillation

Still, when the teacher model is black-box and has top-1 prediction only (i.e., with simple gradient-detaching and argmax operations), the data-free methods are challenged. The earliest to address BBDFKD is ZSDB3 by Wang (2021), which assumes: **(1)** the distance from an image sample to the decision boundary with other classes can represent its robustness, and **(2)** realistic images tend to be far away from these boundaries. This method uses random pixel-wise uniform noises for initialization and optimization. However, such high-dimensional boundaries can be over-complex for distance-based approaches, so the odds of obtaining diverse image samples are extremely slim and scale exponentially worse with high-resolution images or numerous classes. ZSDB3 was only successful in small-size, ten-class datasets.

Two subsequent works, IDEAL by Zhang et al. (2022) and DFHL-RS by Yuan et al. (2024), utilize an extra generator network for more flexible data generation. Both assume that as the student improves during KD, its gradients could guide generator training. The methods employ in-class similarity and class-balance objectives for the generator, which is popular in data-free KD (Chen et al., 2019). However, in black-box scenarios, while distilling an immature student is already challenging, its labels might not be reliable enough to guide generator training. This setup has been observed to cause *"overfitting of synthetic data"* to the student, resulting in possible mode collapse and insufficient diversity (Yuan et al., 2024).

# 3 Proposed Framework

## 3.1 Problem setup

**Problem statement.** Given solely a *black-box* pre-trained teacher network $T$ that returns only top-1 hard labels (no internal features, logits, or gradients) and *no real training data*, our goal is to train a student network $S$ to approximate $T$.

Lacking a direct solution, we may tweak the standard KD framework (Hinton et al., 2015) as a naive solution. Without data, we may resort to using noise images, e.g., $x \sim \mathcal{X}_\mathcal{U}$, where $\mathcal{X}_\mathcal{U} = \mathcal{U}\left[0,1\right]^{C \times H \times W}$ and $C \times H \times W$ is the image dimension. Moreover, we can only use a hard label $\hat{y}_{T-\mathrm{Hard}} = T(x) \in \{1, \ldots, K\}$ to match the student's probabilistic outputs $\hat{y}_{S-\mathrm{Soft}} = S(x)$ with the objective:

$$\mathcal{L}_{\mathrm{Naive\text{-}KD}} = \mathbb{E}_{x \sim \mathcal{X}_\mathcal{U}}\left[\mathcal{L}_{\mathrm{CE}}\left(\hat{y}_{S-\mathrm{Soft}}, \hat{y}_{T-\mathrm{Hard}}\right)\right], \tag{1}$$

where $\mathcal{L}_{\mathrm{CE}}$ is the cross-entropy loss. While this approach can handle simple cases, it is apparent that it fails to use inter-class knowledge — the pivotal advantage in KD. Moreoever, we will show that noise images are semantically not diverse, and that diversity can drive KD performance through our proposed framework.

**Proposal.** To improve image diversity in BBDFKD, we propose *Diverse Image Priors Knowledge Distillation* (**DIP-KD**) of three phases: Synthesis, Contrast, and Distillation.

1. In the **Synthesis** phase, we design a pipeline to create *image priors* $\mathcal{X}_\mathcal{P}$ — synthetic images to capture diversity in hierarchical structures, nonlinearity, and meaningful semantics; which are often observed in natural scenes and objects. We achieve this via three sub-components: hierarchical noise, nonlinear transformation, and semantic cutmixing.

2. Then, in the **Contrast** phase, we sample a dataset $\mathcal{D} := \{x | x \sim \mathcal{X}_{\mathcal{P}}\}$ and optimize them for diversity. We use a contrastive-based objective to make each and every sample *distinguishable*.

3. Finally, in the **Distillation** phase, we use $\mathcal{D}$ to distill the student $S$ from the teacher $T$. Notably, we are able to extract logits from a primer student $S_0$ for soft knowledge to boost KD performance. This is absent in previous BBDFKD methods.

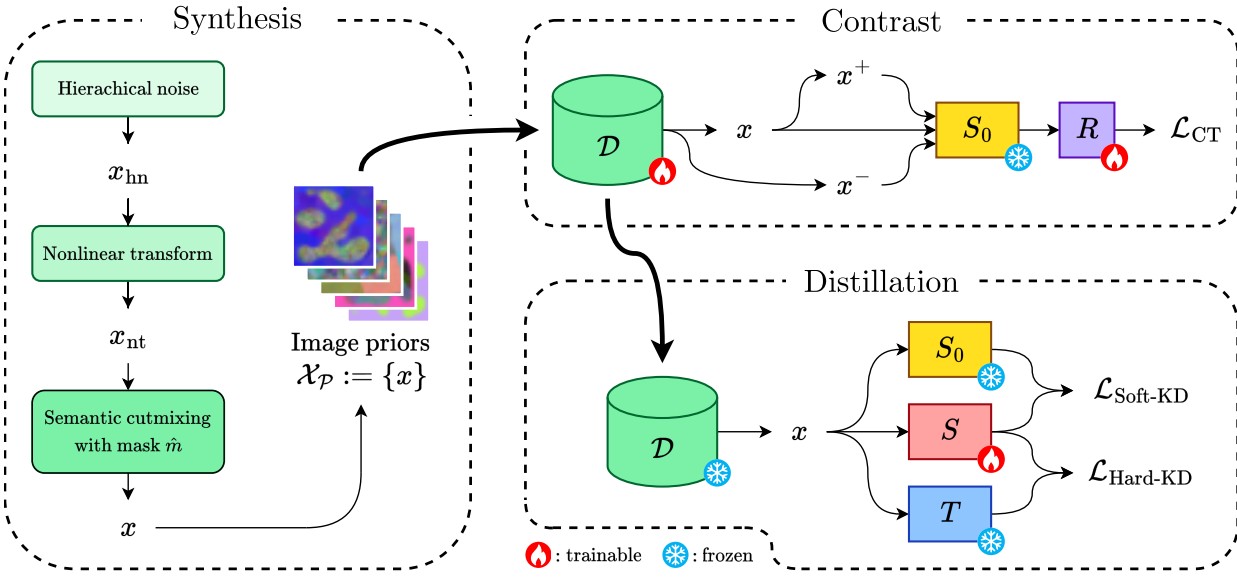

Figure 1: Illustration of our method **DIP-KD** of three phases. **(1) Synthesis:** To create image priors $\mathcal{X}_{\mathcal{P}}$, we create a pipeline of three sub-components: hierarchical noise, nonlinear transformation, and semantic cutmixing. Synthetic images $x \sim \mathcal{X}_{\mathcal{P}}$ have diverse and distinct patterns. **(2) Contrast:** We construct the dataset $\mathcal{D} := \{x | x \sim \mathcal{X}_{\mathcal{P}}\}$ to train an primer student $S_0$ as a feature extractor. Then, we enforce an instance-discriminator $R$ to distinguish embeddings of images $x \sim \mathcal{D}$ to be similar to its positive view $x^+$, and dissimilar to its negative view $x^- \neq x$. Thus, $\mathcal{D}$ is optimized to be even more diverse. **(3) Distillation:** We distill the student $S$ by matching with both hard labels from teacher $T$ via $\mathcal{L}_{\text{Hard-KD}}$ and soft labels from primer student $S_0$ via $\mathcal{L}_{\text{Soft-KD}}$, providing extra knowledge.

## 3.2 Synthesis: crafting diverse image priors

From a visual recognition point-of-view, there exists *strong local structures* between neighbor pixels that compose *global patterns* in natural scenes and objects (LeCun et al., 1998). Moreover, natural objects usually consist of strong nonlinearity, and can capture complex semantics. Based on these premises, we create image priors $\mathcal{X}_{\mathcal{P}}$ with our Synthesis pipeline of three sub-components: *hierarchical noise*, *nonlinear transformation*, and *semantic cutmixing*.

### 3.2.1 Hierarchical noise

Hierarchy is an universal characteristic transcending natural scenes and objects. For instance, akin to how a vehicle is characterized by its windows, mirrors, wheels, an animal is easily recognizable via its eyes, ears, limbs. Based on this observation, we design a sampling technique that captures image patterns at both the local and global scales to mimic the *hierarchical structures* of visual signals.

For a pure pixel-wise noise image $x \sim \mathcal{X}_{\mathcal{U}} = \mathcal{U}[0,1]^{C \times H \times W}$, the patterns are majorly local and independent. Whereas, a monochromatic image (i.e., single-color) has dominant global correlation. Along this spectrum,

there must exist a trade-off between the local-versus-global property. Inspiredly, we design a sampler composing noise images at multiple scales. Let us sample a collection of noise images with increasing dimensions until they sufficiently envelope $H \times W$:

$$\left\{ x_d | x_d \sim \mathcal{U}\left[0,1\right]^{C \times 2^d \times 2^d} \right\}_{d=0}^{d_{\max}}, \tag{2}$$

where $d_{\max} = \text{ceil}\left(\log_2 \max\left(H, W\right)\right)$ is the sufficient scale. Note that the set of scales is derived deterministically from the image dimension and does not require tuning. For instance, we synthesize a $32 \times 32$ image from scales $\{2^0, 2^1, \ldots, 2^5\}$.

Next, we aim to compose this collection of local-to-global noise to a single image. Thus, we proceed to sample coefficients $\{\alpha_d | \alpha_d \sim \mathcal{N}\left(0,1\right)\}_{d=0}^{d_{\max}}$ and put them through softmax to obtain each $\bar{\alpha}_d = \dfrac{\exp\left(\alpha_d\right)}{\sum_{d'=0}^{d_{\max}} \exp\left(\alpha_{d'}\right)}$. This ensures $\sum_{d=0}^{d_{\max}} \bar{\alpha}_d = 1$ and $\bar{\alpha}_d \in [0,1]$ for a valid combination, which is our *hierarchical noise* image:

$$x_{\text{hn}} = \sum_{d=0}^{d_{\max}} \bar{\alpha}_d \cdot f_{\text{upscale}}\left(x_d; 2^{d_{\max}}\right), \tag{3}$$

where $f_{\text{upscale}}\left(\cdot; 2^{d_{\max}}\right)$ is an upscaling transformation so that noise images $\{x_d\}$ share equal dimensions to allow additivity.

### 3.2.2 Nonlinear transformation

To enrich nonlinear patterns of synthetic images, we apply random rotation $f_{\text{rot}}$ of $[-45°, 45°]$, random elastic transform $f_{\text{elas}}$, and random cropping $f_{\text{crop}}$ to the specified size $H \times W$. These operations are known to be nonlinear and effective in data augmentation (Simard et al., 2003), and in our case, boosting data diverity. Accordingly, we create a *nonlinear-transformed* image as:

$$x_{\text{nt}} = f_{\text{crop}}^{H \times W} \circ f_{\text{elas}} \circ f_{\text{rot}}^{[-45°, 45°]}\left(x_{\text{hn}}\right). \tag{4}$$

### 3.2.3 Semantic cutmixing

Finally, the semantics of natural objects usually consist of distinct cohesive shapes as the 'content', distinguishable to the background. Inspired by CutMix (Yun et al., 2019), we propose an improved masking technique to nonlinearly cutmix any two images together, creating diverse and semantically cohesive shapes.

We simulate these patterns within a semantic mask $m$, initialized to $m_0 \sim \mathcal{U}\left[0,1\right]^{1 \times H \times W}$. Regarding where $m < \beta$ as negative regions and $m \geq \beta$ as positive regions, the key to creating structurally cohesive shapes is to connect same-type regions but contrast opposite-type ones. While the former can be easily executed with a blurring filter $f_{\text{blur}}$, we design the diverging filter $f_{\text{divg}}$ for mask $m$ within the $[0,1]$ dynamic range as:

$$f_{\text{divg}}\left(m; 0 \leq \beta \leq 1\right) = \begin{cases} \frac{m^2}{\beta} & , m \geq \beta \\ \frac{m^2 - 2m + \beta}{\beta - 1} & , m < \beta \end{cases}. \tag{5}$$

We provide detailed formulation of $f_{\text{divg}}$ in Appendix Sec. A.3.1. By design, $f_{\text{divg}}$ is composed of two continuously differentiable half-parabolas: the left-side convex one on $[0, \beta)$, the right-side concave on $[\beta, 1]$, connected at the inflection point $(\beta, \beta)$. We fix $\beta = 0.5$ for a balanced formation of positive/negative regions in the masks $\hat{m}$ to keep our method generalizable and interpretable.

To obtain the final mask $\hat{m}$, we iteratively pass mask $m_0$ through the two filters as $m_{i+1} = f_{\text{blur}} \circ f_{\text{divg}}\left(m_i; \beta\right)$ for $M = 10$ times. We then employ $\hat{m}$ to pairwisely cutmix $\{x_{\text{nt}}\}$ against themselves and random monochromatic (i.e., single-color) images $\mathcal{X}_{\text{mc}}$ to form our final *image priors* $\mathcal{X}_{\mathcal{P}} := \{x\}$, where:

$$x = \text{CutMix}\left(x_i, x_j; \hat{m}\right), \tag{6}$$

such that $x_i, x_j \sim \{x_{\text{nt}}\} \cup \mathcal{X}_{\text{mc}}$ and $x_i \neq x_j$. In short, while the randomness in $m_0$ yields diverse initializations, the filters turn them in cohesive shapes in $\hat{m}$ for cutmixing. We showcase how diversity in image priors is enriched throughout the pipeline in Fig. 2, and will later examine their semantics in our analysis in Sec. 4.4.2.

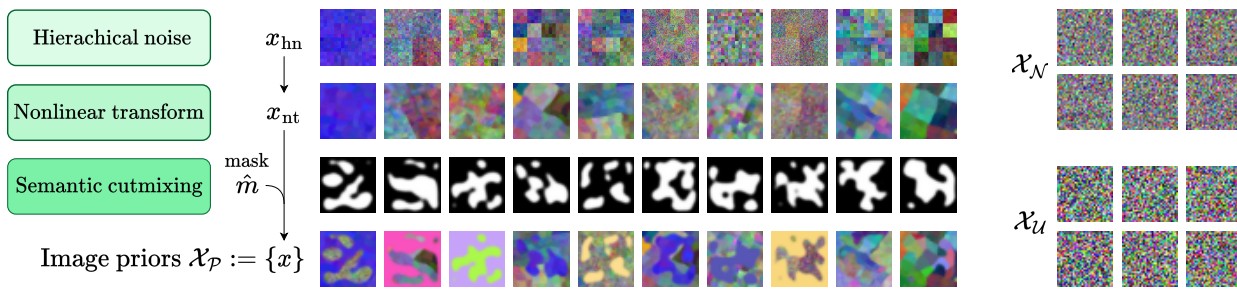

Figure 2: In the **Synthesis** phase, we generate image priors $\mathcal{X}_\mathcal{P}$ with 3 sub-components: **(1)** Hierchical noise: We sample $x_{\text{hn}}$ from multi-scale noise images to capture hierarchical structures; **(2)** Nonlinear transform: We transform $x_{\text{hn}}$ to $x_{\text{nt}}$ with nonlinear filters to capture nonlinearity; and **(3)** Semantic cutmixing: We apply cutmixing on $x_{\text{nt}}$ with a semantic mask $\hat{m}$ to construct the final images $x$, simulating diverse semantics. Observably, image priors $\mathcal{X}_\mathcal{P}$ have superior diversity to noise images sampled from random Gaussian distribution $\mathcal{X}_\mathcal{N}$ and random uniform distribution $\mathcal{X}_\mathcal{U}$.

## 3.3 Contrast: improving diversity via contrastive learning

We further improve the diversity of image priors $\mathcal{X}_\mathcal{P}$ through contrastive learning. However, contrastive methods usually requires extracting intermediate features, which is not feasible from a black-box teacher. Alternatively, we simply train a *primer* student $S_0$ on an image priors dataset $\mathcal{D} = \{x | x \sim \mathcal{X}_\mathcal{P}\}$. We also attempt to make $\mathcal{D}$ class-balanced via rejection sampling, which is efficient as $\mathcal{X}_\mathcal{P}$ is diverse. We train $S_0$ on $\mathcal{D}$ following the Hard-KD objective:

$$\mathcal{L}_{\text{Hard-KD}} = \mathbb{E}_{x \sim \mathcal{D}} \left[ \mathcal{L}_{\text{CE}} \left( \hat{y}_{S_0 - \text{Soft}}, \hat{y}_{T - \text{Hard}} \right) \right], \tag{7}$$

where $\hat{y}_{S_0 - \text{Soft}} = S_0(x)$ is the probability prediction of the primer student, and $\hat{y}_{T - \text{Hard}} = T(x)$ is the hard label queried from the teacher. Equipped with a white-box $S_0$, we can explore its internal representations for contrastive learning.

Following Fang et al. (2021), diversity boils down to *"how distinguishable are the samples from the dataset"*. Interestingly, it can be explicitly formulated as an optimizable objective to mitigate the mode collapse and boost KD performance. For this goal, we utilize the primer student's backbone $S_0^\mathcal{H}$ to extract image embeddings and append an extra *instance-discriminator* $R$ to treat each embedding as a distinct class. The network $R$ is a shallow MLP (two linear layers) to project the general-purpose embeddings to a contrastive-specific space. We argue that even if $S_0$ might not have comparable performance to $T$, its backbone $S_0^\mathcal{H}$ can well capture the semantics, so $R$ distinguishes them effectively. This is supported by the setup in popular contrastive learning frameworks (Chen et al., 2020; He et al., 2020; Zbontar et al., 2021): contrastive representation learning does not depend on a high-accuracy classifier backbone, but often train from randomly-initialized networks. For any two images $(x_i, x_j)$, we chain them through $S_0^\mathcal{H}$ and $R$ to obtain their embeddings and quantify their cosine similarity as:

$$\text{Sim} (x_i, x_j) = \frac{\langle R \circ S_0^\mathcal{H}(x_i), R \circ S_0^\mathcal{H}(x_j) \rangle}{\left\| R \circ S_0^\mathcal{H}(x_i) \right\| \cdot \left\| R \circ S_0^\mathcal{H}(x_j) \right\|}. \tag{8}$$

To enforce contrastive learning, from an image $x \sim \mathcal{D}$, we create positive views $x^+$ with random augmentation and sample different images $x^- \sim \mathcal{D}$ (where $x^- \neq x$) as negative views. The contrastive loss is formulated to maximize the similarity between $(x, x^+)$ and minimize that between $(x, x^-)$:

$$\mathcal{L}_{\text{CT}} = \mathbb{E}_{x \sim \mathcal{D}} \left[ -\log \frac{\exp \left( \text{Sim}(x, x^+) \right)}{\sum_{x^- \neq x} \exp \left( \text{Sim}(x, x^-) \right)} \right]. \tag{9}$$

We parameterized $\mathcal{D}$ and let the gradients of $\mathcal{L}_{\text{CT}}$ flow back to update a batch of $x \in \mathcal{D}$ and the instance-discriminator $R$ at each time step. As the better $R$ can learn to discriminate, the more distinguishable updates each $x \in \mathcal{D}$ can receive, and vice versa. This reinforcement effectively improves the diversity of $\mathcal{D}$.

### 3.4 Distillation: Training the student network

After $\mathcal{D}$ has been optimized for diveristy, we use them to distill the student $S$. We aim to encorporate both *hard* and *soft* knowledge, which is **fundamental** in KD (Hinton et al., 2015), to construct an *optimal* distillation objective. This is either *limited* or *entirely absent* in Wang (2021); Zhang et al. (2022); Yuan et al. (2024). Contrarily, having privilege access to the primer network $S_0$, we can jointly match our student's logits $S^{\mathcal{Z}}(x)$ with logits of the primer student $S_0^{\mathcal{Z}}(x)$ as soft labels, and with queries $\hat{y}_{T-\text{Hard}}$ from the teacher $T$ as hard labels:

$$\mathcal{L}_S = \mathcal{L}_{\text{Hard-KD}} + \mathcal{L}_{\text{Soft-KD}} = \mathbb{E}_{x \sim \mathcal{D}} \Big[ \mathcal{L}_{\text{CE}}(\hat{y}_{S-\text{Soft}}, \hat{y}_{T-\text{Hard}}) + \mathcal{L}_{\text{L1}}\left(S^{\mathcal{Z}}(x), S_0^{\mathcal{Z}}(x)\right) \Big], \qquad (10)$$

where $\mathcal{L}_{\text{L1}}$ is the L1 divergence loss that captures the privilege inter-class knowledge. To conclude our framework, we summarize it as pseudocode in Alg. 1.

---

**Algorithm 1:** Proposed method **DIP-KD**

---

**Input:** a pre-trained teacher network $T$
**Parameters:** number of synthetic images $N$
**Output:** a student network $S$

---

   ▷ *Synthesis*
**1** Construct hierarchical noise images $\{x_{\text{hn}}\}$ via Eq. 2, 3
**2** Apply nonlinear transforms on $\{x_{\text{nt}}\}$ to obtain $\{x_{\text{hn}}\}$ via Eq. 4
**3** Apply semantic cutmixing on $\{x_{\text{hn}}\}$ to obtain $\mathcal{X}_{\mathcal{P}} \coloneqq \{x\}$ via Eq. 5, 6

   ▷ *Contrast*
**4** Construct and parameterize dataset $\mathcal{D} = \{x | x \sim \mathcal{X}_{\mathcal{P}}\}_{i=1}^{N}$
**5** Train primer student $S_0$ by querying $T$ on $\mathcal{D}$ to minimize $\mathcal{L}_{\text{Hard-KD}}$ via Eq. 7
**6** Initialize instance-discriminator $R$
**7** **while** *($\mathcal{D}$ not converged)* **do**
**8**      Sample $x \sim \mathcal{D}$ and their positive $x^+$ and negative views $x^- \neq x$
**9**      Update $x$ and $R$ to minimize $\mathcal{L}_{\text{CT}}$ via Eq. 9

   ▷ *Distillation*
**10** Distill student $S$ by querying $T$ and $S_0$ on $\mathcal{D}$ to minimize via $\mathcal{L}_S$ Eq. 10

**11** **return** $S$

---

## 4 Experiments

To demonstrate the effectiveness of our method DIP-KD, we evaluated our method on extensive BBDFKD benchmarks. Furthermore, we conduct a wide range of ablation studies and analysis to reveal the underlying mechanics of our framework, and validate it under practical scenarios. We also provide training details, complementary studies, and extended works in the Appendix.

### 4.1 Experimental setup

We evaluate our method across eight benchmark datasets at various difficulty: MNIST (LeCun et al., 1998), USPS (Hull, 1994), SVHN (Netzer et al., 2011), FMNIST (Xiao et al., 2017), CIFAR10, CIFAR100 (Krizhevsky et al., 2009), Tiny-ImageNet (Le & Yang, 2015) (abbreviated: TinyImageNet), Imagenette (FastAI, 2020). Three network architectures are considered for the teacher and student, namely LeNet5 (LeCun et al., 1998), AlexNet (Krizhevsky et al., 2012), and ResNet (He et al., 2016). These datasets and architectures are the de facto evaluation in BBDFKD methods (Wang, 2021; Zhang et al., 2022; Yuan et al., 2024). Our proposed hyperparameters $(\beta, M)$ are kept consistent across all experiments, and are not tuned to any datasets, adhering to the data-free constraint. For detailed training setups, we kindly refer readers to Appendix Sec. A.

## 4.2   Baselines

We compare our method DIP-KD with the baselines:

- Naive-KD: The student is trained on uniform noise $x \sim \mathcal{X}_{\mathcal{U}}$, where $\mathcal{X}_{\mathcal{U}} = \mathcal{U}[0,1]^{C \times H \times W}$ and the teacher's hard labels $\hat{y}_{T-\mathrm{Hard}} = T(x) \in \{1, \ldots, K\}$.

- BBDFKD methods: We compare ours with three state-of-the-art methods: ZSDB3 (Wang, 2021), IDEAL (Zhang et al., 2022), and DFHL-RS (Yuan et al., 2024).

For a fair comparisons, we used the same teacher-student network architectures and synthetic images budget for all methods. We report the *mean ± standard error* of distillation accuracy, which is student's accuracy on the hold-out test set, across five runs. The accuracy of baseline BBDFKD methods are reproduced with their official implementations.[1] We ensured that all baseline implementations were tuned ethically and fairly, following the authors' recommended settings.

## 4.3   Standard experiments

We categorize the datasets into *simple* and *complex* datasets based on their difficulty. Four simple datasets MNIST, USPS, SVHN, FMNIST all have 10 classes of simple objects, at a $32 \times 32$ or smaller resolution. The complex datasets have more complicated objects or more classes: CIFAR10 and CIFAR100 (10 and 100 classes, $32 \times 32$), TinyImageNet (200 classes, $64 \times 64$), and Imagenette (10 classes, but high-resolution). More details on the dataset can be found in Appendix Sec. A.1.

**Simple datasets.** We report knowledge distillation (KD) results on simple datasets in Table 1. Across all cases, BBDFKD baselines consistently outperform Naive-KD, which only probes the teacher with noisy pixels. Notably, we observe a consistent accuracy margin between our method DIP-KD and other BBDFKD baselines. This demonstrates the effectiveness of DIP-KD to achieve superior distillation performance.

Table 1: Distillation accuracy (%) on simple datasets.

| Baseline | MNIST LeNet5 $N = 20$ K | USPS LeNet5 $N = 50$ K | SVHN AlexNet $N = 50$ K | FMNIST LeNet5 $N = 50$ K |
|---|---|---|---|---|
| Teacher | 99.28 | 95.47 | 96.16 | 90.90 |
| Naive-KD | $96.54_{\pm 0.60}$ | $42.05_{\pm 0.08}$ | $83.35_{\pm 0.57}$ | $55.11_{\pm 1.23}$ |
| ZSDB3 | $96.66_{\pm 0.39}$ | $65.50_{\pm 0.16}$ | $85.38_{\pm 1.69}$ | $71.60_{\pm 2.35}$ |
| IDEAL | $96.96_{\pm 0.34}$ | $92.66_{\pm 0.57}$ | $87.86_{\pm 0.93}$ | $82.84_{\pm 1.22}$ |
| DFHL-RS | $98.53_{\pm 0.10}$ | $93.97_{\pm 0.60}$ | $95.41_{\pm 0.13}$ | $83.89_{\pm 1.35}$ |
| **DIP-KD** | $\mathbf{98.59}_{\pm 0.08}$ | $\mathbf{94.20}_{\pm 0.21}$ | $\mathbf{95.94}_{\pm 0.05}$ | $\mathbf{83.98}_{\pm 0.65}$ |

$N$: synthetic dataset size; Subscript $_{\pm \ldots}$ denotes the standard error; Best results are in **bold**.

**Complex datasets.** For complex datasets, we summarized distillation performance in Table 2. We observe the relative gap between the baselines to be wider compared to that on simple datasets in Table 1, indicating that these datasets are indeed more difficult to distill. In class-abundant scenarios such as CIFAR100 and TinyImageNet, Naive-KD and ZSDB3 drop to near random-guessing levels, due to the limited diversity in its training data (both used pixel-based noise images). The remaining BBDFKD methods are more robust on these datasets, especially DFHL-RS and our method DIP-KD. However, we notice a significant margin of more than 20% between our method and DFHL-RS, the second-best baseline, on the high-resolution Imagenette. In summary, DIP-KD consistently outperforms on 7/8 baselines with state-of-the-art results, and remains robust under class-abundant or high-resolution settings. These results imply our method DIP-KD has great potential in practical, real-world black-box data-free scenarios.

---

[1]Official implementations of baseline methods: (1) ZSDB3 at `https://github.com/zwang84/zsdb3kd`; (2) IDEAL at `https://github.com/SonyResearch/IDEAL`; (3) DFHL-RS at `https://github.com/LetheSec/DFHL-RS-Attack`.

Table 2: Distillation accuracy (%) on complex datasets.

| Baseline | CIFAR10 ResNet18 $N = 50K$ | CIFAR100 ResNet18 $N = 100K$ | TinyImageNet ResNet18 $N = 512K$ | Imagenette ResNet18 $N = 50K$ |
|---|---|---|---|---|
| Teacher | 95.21 | 78.36 | 66.08 | 91.29 |
| Naive-KD | $13.99_{\pm 0.23}$ | $2.68_{\pm 0.13}$ | $0.75_{\pm 0.07}$ | $33.21_{\pm 0.39}$ |
| ZSDB3 | $56.74_{\pm 1.98}$ | $9.19_{\pm 0.39}$ | $3.29_{\pm 0.16}$ | $33.43_{\pm 0.85}$ |
| IDEAL | $67.58_{\pm 1.29}$ | $33.64_{\pm 0.91}$ | $23.76_{\pm 1.03}$ | $36.90_{\pm 0.38}$ |
| DFHL-RS | $76.50_{\pm 0.84}$ | $51.97_{\pm 0.11}$ | $\mathbf{30.52}_{\pm 1.43}$ | $40.53_{\pm 2.36}$ |
| **DIP-KD** | $\mathbf{83.41}_{\pm 0.46}$ | $\mathbf{52.85}_{\pm 0.30}$ | $28.03_{\pm 0.22}$ | $\mathbf{61.84}_{\pm 0.77}$ |

$N$: synthetic dataset size; Subscript $_{\pm ...}$ denotes the standard error;
Best results are in **bold**.

## 4.4 Ablation studies

In this section, we provide deeper insights into our framework by dissecting the components of our framework, and then put it to practical settings. Formerly, we perform an analysis on individual component's contribution, explore the semantics and visualization of our image priors, and examine our method's sensitivity to hyperparameters. Then, we extend our evaluation to real-world scenarios such as KD with proxy data, with cross-architecture settings, or under aggressive model compression. We also refer enthusiastic readers to visit Appendix Sec. B for complementary studies on alternative optimization objectives, and Appendix Sec. C for an iterative extension of our method.

### 4.4.1 Contribution of components

Table 3: Distillation accuracy (%) by inclusion of components.

| Method | MNIST | CIFAR10 |
|---|---|---|
| Naive-KD (noise data $x \sim \mathcal{X}_{\mathcal{U}}$, hard labels) | $96.02_{\pm .60}$ | $13.99_{\pm .23}$ |
| + **Synthesis:** | + 2.27 | + 64.18 |
| ▲Hierarchical noise: $\mathcal{D} \coloneqq \{x_{\mathrm{hn}}\}$ | $1.79$▲ | $59.26$▲ |
| ▲Nonlinear transform: $\mathcal{D} \coloneqq \{x_{\mathrm{nt}}\}$ | $0.18$▲ | $2.38$▲ |
| ▲Semantic cutmixing: $\mathcal{D} \coloneqq \{x \mid x \sim \mathcal{X}_{\mathcal{P}}\}$ | $0.30$▲ | $2.54$▲ |
| + **Contrast:** Update $\mathcal{D}$ to minimize $\mathcal{L}_{\mathrm{CT}}$ via Eq. 9 | + 0.16 | + 3.14 |
| + **Distillation:** Add Soft-KD to $\mathcal{L}_S$ in Eq. 10 | + 0.14 | + 2.10 |
| Complete **DIP-KD** | $98.59_{\pm .08}$ | $83.41_{\pm .46}$ |

▲ denotes individual contribution of each Synthesis sub-components.
Subscript $_{\pm ...}$ denotes the standard error at the base and complete case.

We investigate each component of our framework DIP-KD on MNIST and CIFAR10 from the standard experiment, as representative for simple and complex benchmarks. Starting from the baseline Naive-KD, we sequentially add each component and report the student accuracy in Table 3. Firstly, Synthesis contributes the most to student accuracy (+2.27% on MNIST, +64.18% on CIFAR10), where the hierarchical noise sub-component has the most impact. This is equivalent to the performance of the primer student $S_0$ trained on $\mathcal{X}_{\mathcal{P}}$, of more than 98% and 78% on MNIST and CIFAR10, which is already higher than the second-best baseline DFHL-RS in the standard experiments (Sec. 4.3).

High-performance primer students allows us to conduct contrastive optimization and perform soft KD effectively. For instance, on CIFAR10, we achieve +3.14% for Contrast and +2.10% for Distillation. These results confirms that not only our Synthesis pipeline plays a vital role in generating diverse data, but also our novel use of the primer student in the Contrast and Distillation phases significantly boosts KD performance beyond prior arts.

### 4.4.2 Embeddings analysis — Generation

To ensure that our image priors are diverse and semantically meaningful, we sample random synthetic images and extract their embeddings from the backbone of the ResNet18 teacher network pre-trained on CIFAR10.[2] Its intermediate features have a dimension of 512, where we further embed with UMAP (McInnes et al., 2018) with default parameters to two-dimensional for visualization. We explore four complex datasets CIFAR10, CIFAR100, TinyImageNet, and Imagenette, which contain rich diversity and semantics. We declare that the use of real data and the teacher's backbone in this ablation study is solely for evaluation purpose, and is not involved in our framework. We also include uniform noise $\mathcal{X}_{\mathcal{U}}$, Gaussian noise $\mathcal{X}_{\mathcal{N}}$, and our image priors $\mathcal{X}_{\mathcal{P}}$. We visualize these embeddings in Fig. 3 for a qualitative study.

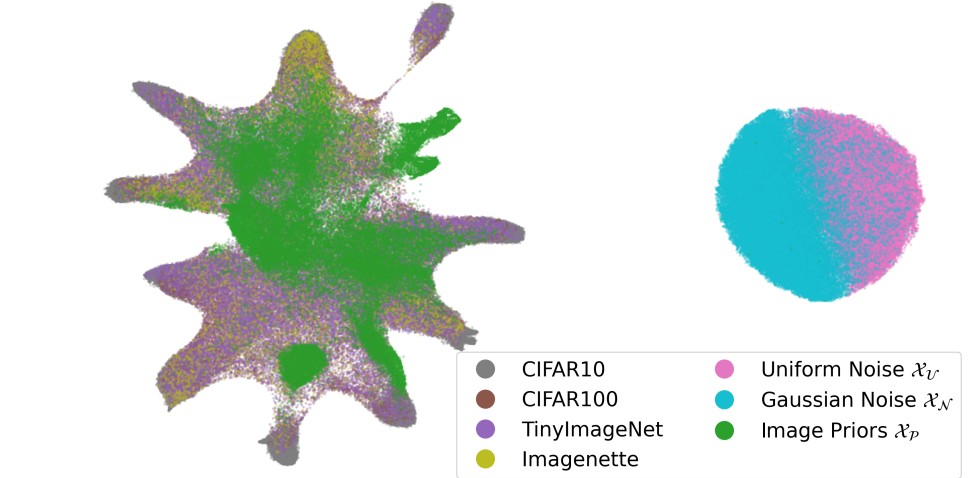

Figure 3: Feature embeddings of complex datasets, noise ($\mathcal{X}_{\mathcal{U}}$, $\mathcal{X}_{\mathcal{N}}$), and image priors $\mathcal{X}_{\mathcal{P}}$.

Apparently, $\mathcal{X}_{\mathcal{U}}$ and $\mathcal{X}_{\mathcal{N}}$ form a compact, separate cluster far away from the regions of real images, suggesting that their semantics are limited and dissimilar to natural objects. In contrast, $\mathcal{X}_{\mathcal{P}}$ widely spreads among the mutual regions of real images, hinting that they share similar semantics as expected. For quantitative analysis, we evaluate four distribution metrics for generative models from Naeem et al. (2020):

- *Precision*: the fraction of synthetic samples that is in the neighborhood of at least one real sample. High precision indicates synthetic data are captured within the support of real data, but may be *biased* by real outlier samples.

- *Recall*: the fraction of real samples that is in the neighborhood of at least one synthetic sample. High recall indicates synthetic data can capture the support of real data, but may be *biased* by an overestimated synthetic manifold.

- *Density*: the average normalized count of synthetic samples in the neighborhood of each real sample. High density indicates *fidelity* like precision but is less *biased*, and may be greater than 1 when synthetic data concentrates around real modes.

- *Coverage*: the fraction of real samples whose neighbourhoods contain at least one fake sample. High coverage indicates *diversity* like recall but is less *biased*.

The statistics in Table 4 shows that our image priors $\mathcal{X}_{\mathcal{P}}$ achieved significantly higher density, coverage, and recall at a small trade-off in precision, compared to noise sources $\mathcal{X}_{\mathcal{U}}$ and $\mathcal{X}_{\mathcal{N}}$. Theses results further confirm $\mathcal{X}_{\mathcal{P}}$ have superior fidelity and diversity, and explains why $\mathcal{X}_{\mathcal{P}}$ excels in BBDFKD.

---

[2]This is the ResNet18 teacher of 95.21% accuracy on CIFAR10, being reused as a held-out "oracle".

Table 4: Distribution metrics (%) on complex datasets.

| Metric | Source | CIFAR10 | CIFAR100 | TinyImageNet | Imagenette |
|--------|--------|---------|----------|--------------|------------|
| **Density** | $\mathcal{X}_{\mathcal{U}}$ | 20.00 | 100.65 | 108.65 | 101.20 |
| | $\mathcal{X}_{\mathcal{N}}$ | 20.00 | 107.98 | 90.87 | 103.64 |
| | $\mathcal{X}_{\mathcal{P}}$ | 49.14 | 96.20 | 96.00 | 92.14 |
| **Coverage** | $\mathcal{X}_{\mathcal{U}}$ | .00 | .10 | .06 | .05 |
| | $\mathcal{X}_{\mathcal{N}}$ | .00 | .10 | .06 | .05 |
| | $\mathcal{X}_{\mathcal{P}}$ | 7.33 | 60.79 | 63.88 | 78.48 |
| **Precision** | $\mathcal{X}_{\mathcal{U}}$ | 100.00 | 100.10 | 100.00 | 100.00 |
| | $\mathcal{X}_{\mathcal{N}}$ | 100.00 | 100.10 | 100.00 | 100.00 |
| | $\mathcal{X}_{\mathcal{P}}$ | 80.83 | 97.76 | 99.29 | 99.10 |
| **Recall** | $\mathcal{X}_{\mathcal{U}}$ | .00 | .09 | .06 | .04 |
| | $\mathcal{X}_{\mathcal{N}}$ | .00 | .10 | .06 | .05 |
| | $\mathcal{X}_{\mathcal{P}}$ | 66.92 | 96.75 | 96.86 | 95.07 |

### 4.4.3 Embeddings analysis — Contrast

To highlight the effect of the Contrast phase on increasing diversity of image priors, we visualize the embeddings of 50K images before and after the Contrast phase in Fig. 4. Embeddings are extracted similarly as previously described, using the same ResNet18 backbone pre-trained on CIFAR10. As an oracle (has been trained on real CIFAR10 images), the backbone can proficiently recognize the semantics and embed real images into ten distinctive clusters (colored gray). Image priors $\mathcal{X}_{\mathcal{P}}$ before Contrast (green) already scatter around these clusters, indicating they have captured diverse semantics. After Contrast, the contrastive objective further improves the coverage and expands $\mathcal{X}_{\mathcal{P}}$ to neighbor regions as well (orange). This is beneficial to KD, as the post-contrast data distribution is both more diverse and help capture more knowledge from the teacher.

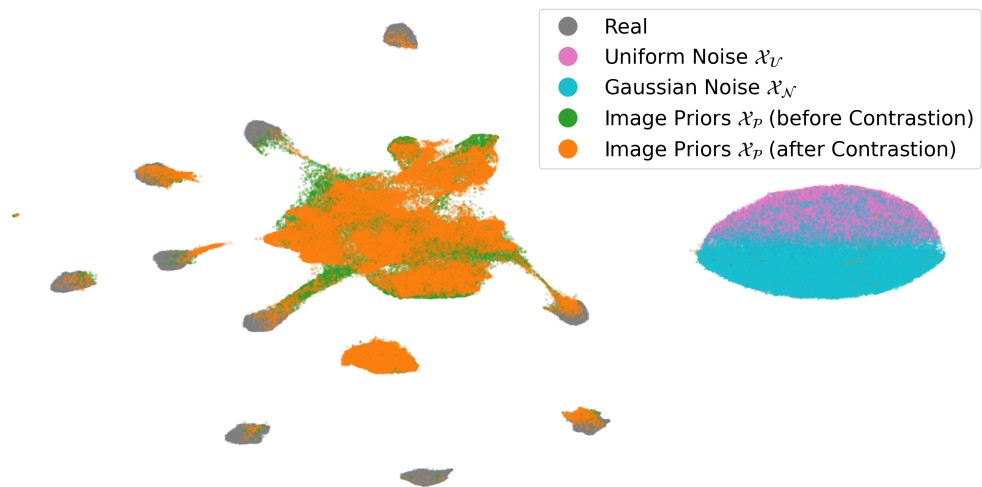

Figure 4: Feature embeddings of image priors $\mathcal{X}_{\mathcal{P}}$ on CIFAR10 before and after the Contrast step, with real data and noise sources.

Upon closer inspection of randomly-picked image priors (Fig. 5), we observe that the contrastive optimization has overlaid the image priors with faint but self-distinguished patterns after Contrast. In most images, the residual can be seen to reinforce the existing patterns. This is expected, as the contrastive loss encourages each image prior to be more distinguishable from others, thus promoting diversity. We hypothesize that these subtle patterns are sufficient for the teacher and the primer student to recognize different semantics, thus improving KD performance.

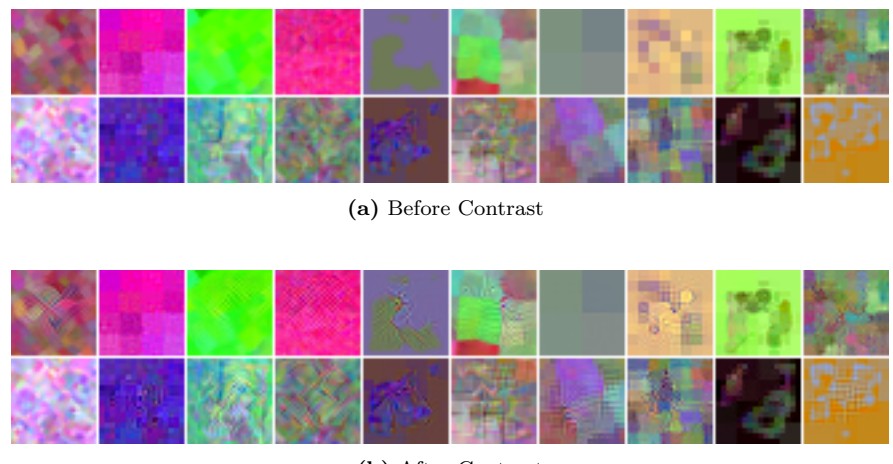

**(a)** Before Contrast

**(b)** After Contrast

Figure 5: Contrastive optimization to image priors $\mathcal{X}_\mathcal{P}$ during the Contrast phase.

Using the same distribution metrics as the previous embedding analysis, we quantitatively evaluate the effect of the Contrast phase on CIFAR10 synthetic data in Table 5. We observe a significant increase in density, coverage, and recall, with a trivial trade-off in density. Overall, these results show that the Contrast phase improve image diversity semantically, visually, and empirically.

Table 5: Distribution metrics (%) during the Contrast phase.

|  | Density | Coverage | Precision | Recall |
|---|---|---|---|---|
| Before Contrast | 70.46 | 6.13 | 91.69 | 82.18 |
| After Contrast | 66.93 | 14.66 | 91.68 | 94.00 |

### 4.4.4   Increasing synthetic dataset size

While it is unsurprising that we achieved better KD performance when using more a larger number of image priors $N$ (Fig. 6), more important discussions are: which is the optimal cut-off point, and how it corresponds to the relative difficulty of each dataset. We observe that for the digits datasets (MNIST, UPSP, SVHN), the student fits easily even with minimal data ($N = 10K$), whereas it struggles a bit more on FMNIST and CIFAR10. For CIFAR100, TinyImageNet, and Imagenette, the student still stably improves even after we have reached $N = 100K$ or more, suggesting that increasing $N$ is a viable option for large-scale problems.

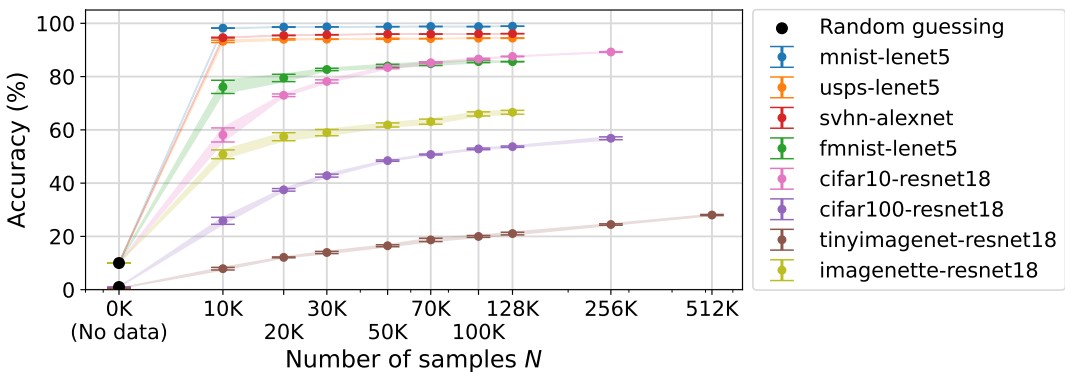

Figure 6: Distillation accuracy (%) by number of image priors $N$.

#### 4.4.5 Distilling with proxy data

Thanks to the digits datasets (MNIST, UPSP, SVHN) having shared classes (digits 0–9), we also explore using out-of-domain datasets as proxy data as another naive approach. We evaluate students trained with different combinations of input images (real training sets, noises, and our image priors) and labels (ground truth from real training sets, or pseudo-labels given by the teacher). We use 50K images for each synthetic source, and the full training set for each real dataset. For fair comparisons, we only include the *Synthesis* component (i.e., accuracy of primer student $S_0$) for our method, and reuse the teachers from standard experiments. We also emphasize that using real training sets containing shared classes is *unfair* to our data-free premise, which starts with strictly no data.

Table 6: Distillation accuracy (%) with proxy data.

| Training data | | Evaluation | | |
|---|---|---|---|---|
| **Inputs** | **Labels** | **MNIST** LeNet5 | **USPS** LeNet5 | **SVHN** AlexNet |
| MNIST | ground-truth | $\mathbf{99.25}_{\pm .06}$ | $91.88_{\pm 1.56}$ | $20.77_{\pm .29}$ |
| | pseudo | $\mathbf{99.06}_{\pm .05}$ | $93.94_{\pm .30}$ | $41.18_{\pm 3.27}$ |
| USPS | ground-truth | $77.92_{\pm 2.72}$ | $\mathbf{95.90}_{\pm .48}$ | $19.36_{\pm .62}$ |
| | pseudo | $96.78_{\pm .42}$ | $\mathbf{94.97}_{\pm .15}$ | $57.52_{\pm 1.81}$ |
| SVHN | ground-truth | $64.10_{\pm 1.29}$ | $62.12_{\pm 1.68}$ | $\mathbf{95.97}_{\pm .08}$ |
| | pseudo | $98.59_{\pm .18}$ | $93.95_{\pm .37}$ | $\mathbf{96.48}_{\pm .04}$ |
| $\mathcal{X}_\mathcal{N}$ | | $88.05_{\pm .76}$ | $27.29_{\pm .10}$ | $84.42_{\pm .48}$ |
| $\mathcal{X}_\mathcal{U}$ | pseudo | $96.54_{\pm .60}$ | $42.05_{\pm .08}$ | $83.35_{\pm .57}$ |
| $\mathcal{X}_\mathcal{P}$ | | $\mathbf{98.62}_{\pm .17}$ | $\mathbf{94.14}_{\pm .52}$ | $\mathbf{95.84}_{\pm .10}$ |

Subscript $_{\pm ...}$ denotes the standard error.
The top-3 results from each target evaluation (column) are in **bold**.

We adopt the perspective of domain adaptation theory in Ben-David et al. (2010) to discuss our results summarized in Table 6.

- First, when the teacher's original training set is available—representing an in-domain scenario—the student model achieves the highest performance. This outcome is expected, as having access to an identical curriculum allows the student to mimic the teacher most effectively.

- Second, performing knowledge distillation in an adaptation-like manner (e.g., from MNIST to USPS) by querying the teacher on proxy datasets provides only a modest improvement over using drop-in labels. Despite the availability of high-quality images, this approach **fails** to achieve satisfactory performance. We hypothesize that this limitation is caused by the domain gap between the teacher's knowledge and the foreign data. Hence, employing public datasets as direct proxies may, in fact, degrade KD performance.

- Finally, rather than *exploitation* over an *uncertain* domain, it is more advantageous to prioritize *exploration* over a *diverse* domain, especially in the BBDFKD setting. Our image priors $\mathcal{X}_\mathcal{P}$ consistently outperform other synthetic sources ($\mathcal{X}_\mathcal{N}$, $\mathcal{X}_\mathcal{U}$) and proxy datasets across all evaluations. Evidently, our results with image priors $\mathcal{X}_\mathcal{P}$ (bottom row) approximates the in-domain settings with less than a 1% difference. This shows that the diversity and broad coverage of our image priors are more beneficial than naively using real data of unsure domain.

Collectively, these insights indicate our image priors exhibits a *universal* diversity and broad applicablity.

#### 4.4.6 Cross-architecture KD

An under-investigated setting in BBDFKD methods is distilling from the teacher to students of heterogeneous architecture. To fully discriminate the knowledge gap, we evaluate three architectures with large capacity

difference: LeNet5, AlexNet, and ResNet18; on three benchmarks with intermediate difficulty: FMNIST, SVHN, and CIFAR10. We reuse the same teacher networks from the standard experiments and report the KD results in Table 7. On the diagonal, we observe that the student performs best when it has a homogeneous architecture to the teacher. Otherwise, accuracy scales with capacity, and a medium-size student can still often give moderate results. Thus, DIP-KD is proven to be applicable even when the teacher and student may have different architectures.

Table 7: Distillation accuracy (%) of cross-architecture KD.

| Network | FMNIST LeNet5 | SVHN AlexNet | CIFAR10 ResNet18 |
|---|---|---|---|
| Teacher | 90.90 | 96.16 | 95.21 |
| LeNet5 | $\mathbf{83.98}_{\pm 0.65}$ | $64.44_{\pm 1.83}$ | $32.01_{\pm 3.18}$ |
| AlexNet | $78.98_{\pm 0.92}$ | $\mathbf{95.94}_{\pm 0.05}$ | $61.30_{\pm 0.69}$ |
| ResNet18 | $81.69_{\pm 1.58}$ | $95.91_{\pm 0.04}$ | $\mathbf{83.41}_{\pm 0.46}$ |

Subscript $_{\pm ...}$ denotes the standard error.
Best results are in **bold**.

### 4.4.7 Students of compressed capacity

While we used same-size teacher-student networks in the standard experiment for fair comparision, we are also interested in the case when the student gets aggressively compressed, which has yet to be investigated in previous methods. Specifically, we train on student networks of approximately 25%, 4%, and 1% compression ratio (CR) by reducing the number of channels in linear and convolutional layers by $2\times$, $5\times$, $10\times$, respectively (Table 8). We distill these compressed students using our method, while keeping all other settings the same as in the standard experiments.

Table 8: Parameters count by compression ratio (CR).

| CR | LeNet5 | AlexNet | ResNet18 |
|---|---|---|---|
| 100% | 61.71 K | 1,659.18 K | 11.17 M |
| 50% | 15.74 K | 417.43 K | 2.90 M |
| 20% | 2.53 K | 68.49 K | 0.45 M |
| 10% | 0.88 K | 17.70 K | 0.11 M |

We report the KD results in Fig. 7 and observe that our method is empirically robust to moderate compression. Even at 25% CR, the students are mostly unaffected across all datasets, and only suffer a tolerable decline in accuracy around 4% CR. Only at the extreme 1% CR, the students starts to decline significantly. This encouraging result shows that DIP-KD is effective in distilling a cumbersome teacher into lightweight students, which is particularly useful for *resource-constrained* settings, such as mobile or edge devices.

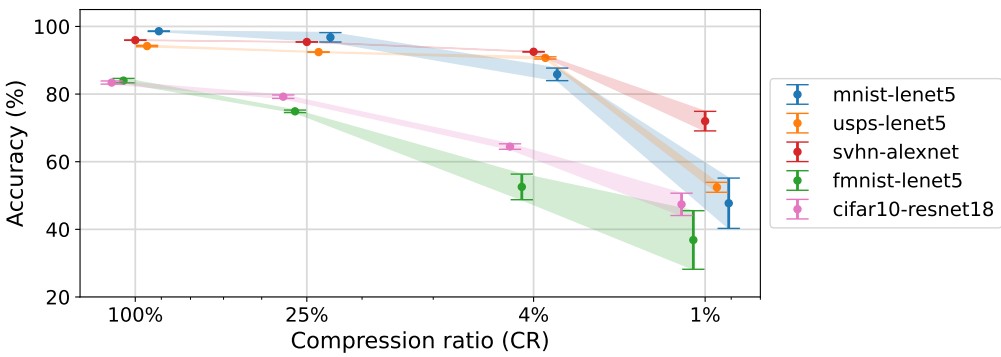

Figure 7: Distillation accuracy (%) by compression ratio (CR).

### 4.4.8 Number of filter iterations in Synthesis

As described in Semantic Cutmixing (Sec. 3.2.3), we generate the semantic mask $\hat{m}$ by iteratively passing an initial noisy mask $m_0 \sim \mathcal{X}_{\mathcal{U}}$ through the filters $f_{\text{blur}} \circ f_{\text{divg}}$ for $M = 10$ times. Following the data-free constraint, we do not tune the number of filter iterations $M$ on any real data, but rather set it based on the visual quality of $\hat{m}$. To investigate its effect, we vary $M$ on a wider spectrum and report the results below.

In particular, we visualize in Fig. 8 the gradual formation of cohesive shapes in semantic masks $\hat{m}$ as $M$ increases. Each pass through the filters transform noisy regions into clearer "blobs", resulting in diverse cohesive shapes. At low $M < 5$, mask $\hat{m} \approx m_0$ is still mostly salt-and-pepper, too fine-grained to capture any cohesive shapes. Meanwhile, at high $M > 20$, the shapes are reduced to simple blobs with less diversity. In practice, the desired data distribution is unknown and may span various domains (e.g., digits, fashion items, animals, objects), so we expect $\hat{m}$ to be general-purpose and robust across domains. We observe a balance between diversity and complexity around $M = 10$, and thus use this value for all our experiments.

Formation of semantic mask $\hat{m}$ at different number of filter iterations $M$

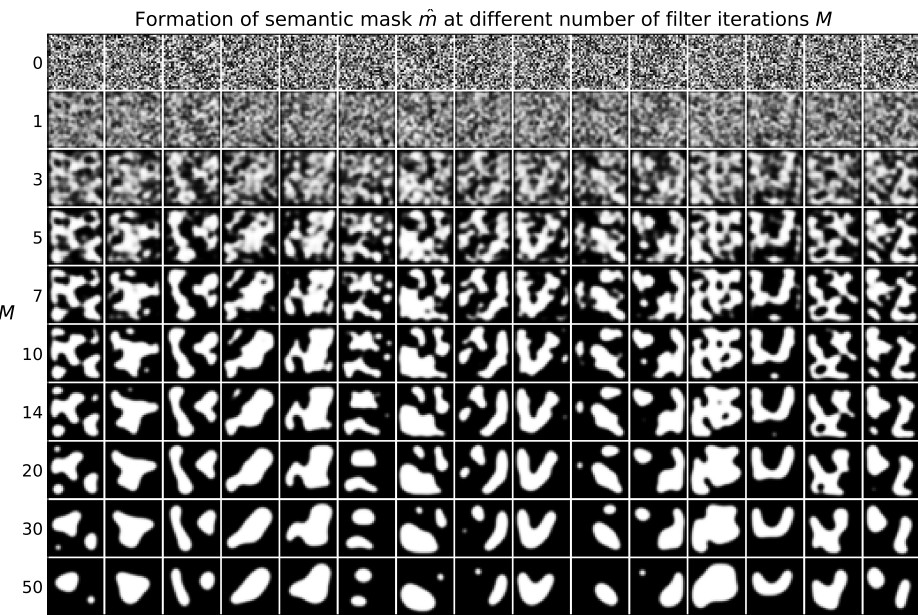

Figure 8: Formation of semantic mask $\hat{m}$ at different number of filter iterations $M \in [0, 50]$.

We additionally experiment on how KD performance varies with $M \in [5, 20]$ in Table 9. As we choose $M$ by generalizability, it is possible that the exact best value of $M$ may differ for each dataset. In our case, our default value $M = 10$ gives us the best overall results, while other values also give robust performance, higher than that of the second-best baseline. We recommend in case where additional priors are available (e.g., access to the domain or dataset), one may opt to tune $M$ to improve an extra 1-2% accuracy margin.

Table 9: Distillation accuracy (%) by number of filter iterations $M$.

| Method | $M$ | MNIST LeNet5 $N = 20$ K | FMNIST LeNet5 $N = 50$ K | CIFAR10 ResNet18 $N = 50$K |
|---|---|---|---|---|
| DIP-KD | 5 | $98.55_{\pm 0.11}$ | $83.57_{\pm 0.13}$ | $81.36_{\pm 0.43}$ |
| | 7 | $98.59_{\pm 0.11}$ | $83.75_{\pm 0.78}$ | $82.29_{\pm 0.29}$ |
| | 10* | $98.59_{\pm 0.08}$ | $83.98_{\pm 0.65}$ | $83.41_{\pm 0.46}$ |
| | 14 | $98.71_{\pm 0.04}$ | $83.80_{\pm 0.30}$ | $82.68_{\pm 0.18}$ |
| | 20 | $98.71_{\pm 0.09}$ | $83.00_{\pm 0.41}$ | $82.31_{\pm 0.72}$ |
| DFHL-RS (second-best) | - | $98.53_{\pm 0.10}$ | $83.89_{\pm 1.35}$ | $76.50_{\pm 0.84}$ |

* denotes our default case using $M = 10$.

## 5 Ethical considerations

Our study explores *black-box data-free knowledge distillation*, a setting where a student model learns from a teacher's outputs without access to its internal parameters or original training data. While not apparent, we would like to raise certain ethical and societal concerns related to model extraction, information leakage, and bias propagation.

Because the student learns entirely from the teacher's responses, repeated querying could approximate the teacher's decision boundaries, potentially enabling unauthorized reconstruction of proprietary or commercial models (Tramer et al., 2016; Orekondy et al., 2019; Jagielski et al., 2020). Large-scale replication of closed models could discourage openness in AI research and reduce incentives for model providers to offer public APIs. This problem could be mitigated by limiting the queries to the teacher, and through model watermarking or output perturbation that make functional copying more detectable or less precise (Jia et al., 2021; Wang et al., 2024).

Besides, even though no real data are used, teacher models may implicitly memorize sensitive information that can be exposed through their responses (Carlini et al., 2021). In extreme cases, this could lead to indirect leakage of personal or confidential data. Such leakage risks eroding public trust in machine learning systems and may breach data protection norms if human-derived models are involved. As a recommendation, using privacy-preserving training techniques, such as differential privacy (Flemings & Annavaram, 2024), and conducting ethical audits of teacher outputs can reduce the likelihood of revealing sensitive patterns.

In data-free KD, the student may be trained using only the teacher's labels, thus it can directly inherit and sometimes amplify distillation-induced biases in the teacher's decision boundaries (Hooker et al., 2020; Ahn et al., 2022). Those amplified disparities can disproportionately harm marginalized groups when distilled models are deployed in downstream settings (Buolamwini & Gebru, 2018). Mitigation strategies include using fairness-aware distillation and audit methods (Chai et al., 2022; Masroor et al., 2024).

Our aim is to enable ethical and resource-efficient research on knowledge distillation, not to facilitate extraction or imitation of closed systems. We therefore restrict our experiments to openly licensed teacher models and open-source our works to encourage transparent, auditable research practices.

## 6 Conclusion

While most knowledge distillation techniques can train competent students, they often struggle in real-world settings where access to the original training data and teacher network is limited. To bridge this gap, we introduce DIP-KD, a novel framework for data-free knowledge distillation from a black-box teacher. Our framework synthesizes diverse images through a tailored generation pipeline, enhance their distinction via contrastive learning, and distill a student using both hard and soft knowledge. We achieve state-of-the-art results on eight benchmarks and deliver comprehensive analyses and experiments. We hope this work not only solves a non-trivial KD problem but also sparks new directions and challenges for future KD research.

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

# Appendix

In this extended text, we provide our implementation details (A), a complementary study on alternative optimization objectives for our image priors (B), and an extension to an iterative framework (C). Explicit results to the figures in the main text are also provided for completeness and reproducibility (D).

## A  Training Details

### A.1  Datasets

In our experiments (Sec. 4), we evaluate our method on eight benchmark image datasets: MNIST (LeCun et al., 1998), USPS (Hull, 1994), SVHN (Netzer et al., 2011), FMNIST (Xiao et al., 2017), CIFAR10, CIFAR100 (Krizhevsky et al., 2009), TinyImageNet (Le & Yang, 2015), Imagenette (FastAI, 2020). All datasets are public and easily downloadable with standard libraries. In terms of image content, MNIST, USPS, and SVHN contain small images of digits. FMNIST is also a small-sized dataset of fashion items. The images in CIFAR10, CIFAR100, TinyImageNet, and Imagenette all contain diverse scenes and objects, e.g., animals, plants, food, vehicles, and household goods, to name a few. Notably, TinyImageNet and Imagenette have higher resolution: $64{\times}64$ for TinyImageNet, and approximately one magnitude larger than that for Imagenette (ranging from $27{\times}80$ to $4368{\times}2912$ — which we reported the average dimension). Of all the benchmarks, only CIFAR100 and TinyImageNet have abundant classes (100 and 200, respectively), while the others have 10 classes. Regarding the dataset size, USPS and Imagenette are relatively smaller. We summarize their details in Table A.1.

Table A.1: Details of the image benchmark datasets.

|         | Dataset | Dimension | Size train | Size test | No. classes | Content |
|---------|---------|-----------|------|------|---------|---------|
| Simple  | MNIST | $1{\times}28{\times}28$ | 60 K | 10 K | 10 | Digits |
|         | USPS | $1{\times}16{\times}16$ | 9 K | 2 K | 10 | Digits |
|         | SVHN | $3{\times}32{\times}32$ | 73 K | 26 K | 10 | Digits |
|         | FMNIST | $1{\times}28{\times}28$ | 60 K | 10 K | 10 | Fashion items |
| Complex | CIFAR10 | $3{\times}32{\times}32$ | 50 K | 10 K | 10 | Diverse |
|         | CIFAR100 | $3{\times}32{\times}32$ | 50 K | 10 K | 100 | Diverse |
|         | TinyImageNet | $3{\times}64{\times}64$ | 100 K | 10 K | 200 | Diverse |
|         | Imagenette | $3{\times}408{\times}472$▲ | 9 K | 4 K | 10 | Diverse |

▲: The average dimension of Imagenette is reported.

### A.2  Teacher and students training

#### A.2.1  Augmentation

To augment the training data (real images for the teacher, synthetic images for students), we design a basic augmentation pipeline. Firstly, if the images are very small (in MNIST, USPS, FMNIST), we resize them to at least $32 \times 32$. Then the images undergo random shifting by $1/8$ of their dimension, and random horizontal flipping unless they are from a digits dataset (MNIST, USPS, SVHN). Finally, we randomly jitter the color in the images for more image variety.

#### A.2.2  Optimization and scheduling

To train the Teacher for baseline, and the Students in our method, we use a stochastic gradient descent optimizer with momentum of 0.9, weight decay of $5 \times 10^{-4}$, and batch size of 100. On datasets with small black-and-white images (MNIST, USPS, FMNIST), we use a fixed learning rate of $10^{-2}$ and train for 100 epochs. On other datasets, we set the learning rate to $10^{-1}$ at the beginning and gradually decrease by two orders of magnitude with a scheduler to enable optimal convergence for 200 training epochs.

### A.3  Optimization of Image Priors

#### A.3.1  Diverging filter

We provide more details on the formulation of the diverging filter $f_{\text{divg}}$ (Eq. 5) proposed in the Synthesis phase (Sec. 3.2). To fulfill its role in contrasting opposite-type regions in a mask $m \in [0,1]^{C \times H \times W}$, we expect $f_{\text{divg}}$ to satisfy the following properties:

1. We start by picking a brightness cutoff threshold $\beta$ to separate the negative $(x < \beta)$ and postive regions $(x \geq \beta)$ for any pixel $x \in m$. The extrema and the threshold value must be fixed after filtering, i.e., $f_{\text{divg}}(\cdot; \beta)$ must go through $\{(0,0), (\beta, \beta), (1,1)\}, \forall \beta$.

2. To preserve the order of intensity across the whole mask, it is necessary that $f_{\text{divg}}$ is an increasing function, i.e., $f'_{\text{divg}}(x; \beta) > 0, \forall x, \beta \in [0,1]$.

3. To avoid creating abrupt changes in intensity, i.e., $f_{\text{divg}}$ must be continuously differentiable on $[0,1], \forall \beta$.

4. To ensure the diverging property emerges at the threshold value $x = \beta$, we require that $f_{\text{divg}}$ has the sharpest increase there, i.e., $f'_{\text{divg}}(\cdot; \beta)$ has a maximum at $x = \beta \Leftrightarrow f''(x = \beta; \beta) = 0, f'(x = \beta; \beta) > 0$.

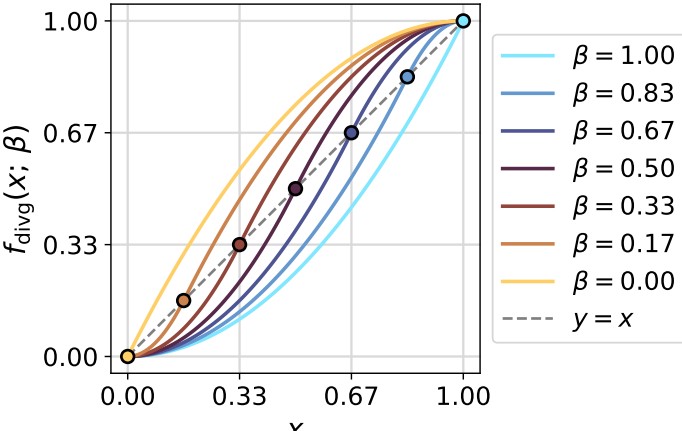

Figure A.1:  Our diverging filter as $y = f_{\text{divg}}(x; \beta)|_{x \in [0,1]}$ in Eq. 5 with different choices of $\beta \in [0,1]$. Scattered points $\{(\beta, \beta)\}$ along the diagonal $y = x$ are also inflection points.

Accordingly, we design a piecewise function consisting of two continuously differentiable half-parabolas connected at the inflection point $(\beta, \beta)$. One of them has a minimum at the $x = 0$ boundary, and vice versa, the other half-parabola has a maximum at the opposite boundary $x = 1$, i.e., $f'_{\text{divg}}(0; \beta) = f'_{\text{divg}}(1; \beta) = 0 \forall \beta \in [0,1]$. Solving the two parabolas with the proposed set of constraints indeed gives the solution we reported $f_{\text{divg}}$ in Eq. 5. We plot $f_{\text{divg}}$ with a range of $\beta \in [0,1]$ values in Fig. A.1 to demonstrate its diverging property. We believe that this filter design has unexplored potentials in image augmentation for robust learning of vision tasks.

#### A.3.2  Loading unique embeddings during Contrast phase

In the Contrast phase, we apply the contrastive method proposed in Fang et al. (2021) to make our data diverse. However, we occasionally observed numerical inbstability and synthetic image corruption during contrastive optimization. After investigation, we found that the issue stems from how embeddings are loaded for contrastive learning. Originally, the embeddings of past inputs $x$ and their corresponding positive views $x^+$ are cached to and recalled from an index-free memory bank. (The same occurred for negative samples,

but it is not linked to the error.) As a result, for small datasets, an embedding loaded from the memory bank might be near-duplicated with one extracted at runtime as, by chance, they may originate from the same image and differ by some trivial updates. Optimization-wise, contrasting an image against itself is counter-intuitive, which explains the exploding gradients and corruption of synthetic images.

Therefore, we modify how images are loaded for contrastive optimization. Instead of using an index-free memory bank, we store the embeddings of each synthetic images in our dataset by its unique index. At contrastive learning runtime, we only load embeddings of images **not** in the current batch, successfully avoid duplication. This modification fixes the numerical instability and synthetic image corruption on our ends.

### A.3.3   Instance-discriminator

Following Fang et al. (2021), we use a fully connected instance-discriminator $R$ comprising of two libear layers. For each input image $x$, $R$ projects the image embedding $h = S_0^{\mathcal{H}}(x)$ from the primer student backbone $S_0^{\mathcal{H}}$, through one hidden layer, to its own embedding space of 128 dimensions. To update $R$ during the Contrast phase (Sec. 3.3), we let the contrastive loss $\mathcal{L}_{\mathrm{CT}}$ in Eq. 9 backpropagate to it parameters. It is trained with an Adam optimizer (Kingma & Ba, 2014) with a learning rate of $10^{-3}$, and its "betas" coefficients of $(0.5, 0.999)$. As $R$ is shallow, this does not incur a significant computational overhead.

### A.3.4   Resources

We perform each standard experiment on a single NVIDIA V100 GPU with 40GB VRAM, 4–16 CPUs, and 4–128 GB RAM, taking around 6–96 hours. The required resources depend on the dataset complexity, model size, and synthetic image resolution. In an increasing order, their training time rank as: simple datasets (negligible difference), CIFAR10, CIFAR100, Imagenette, and TinyImageNet.

## B   Alternative image optimization objectives

In the following section, we will examine student accuracy when adopting different optimization objectives for image prior optimization, and justify our choice of the contrastive approach for diversity. During the Contrast phase, we employ the contrastive loss $\mathcal{L}_{\mathrm{CT}}$ in Eq. 9 to make synthetic images more diverse. Unlike the diversity objective in our framework, previous BBDFKD methods such as Zhang et al. (2022); Yuan et al. (2024) attempted to promote in-class similarity and class-balance. Specifically, in-class similarity is optimized via the one-hot loss:

$$\mathcal{L}_{\mathrm{OH}} = \mathcal{L}_{\mathrm{CE}}\left(\hat{y}_{S-\mathrm{Soft}}, \mathrm{argmax}(\hat{y}_{S-\mathrm{Soft}})\right), \tag{B.1}$$

which guides data generation to strengthen the effect of the most prominent class in each sample. The one-hot loss is popular in data-free KD (Chen et al., 2019), but as stated in Yuan et al. (2024), its adversarial nature might cause "overfitting of synthetic data to the clone model", resulting in mode collapse and insufficient diversity.

The other objective, class-balance, is to ensure the samples are evenly distributed between all classes, enforced by the information entropy loss:

$$\mathcal{L}_{\mathrm{IE}} = \sum_{k=0}^{K-1} p_k \log p_k, \text{ with } p_k = \mathbb{E}_{x \sim \mathcal{D}}\left[\hat{y}_{S-\mathrm{Soft}}^{(k)}\right], \tag{B.2}$$

where $k$ denotes the $k$-th class probability. In our framework, as we have actively balanced the dataset during sampling, class-balance has already been satisfied.

To measure how these objectives might affect KD performance, we ablate them using different losses to optimize the image priors dataset $\mathcal{D} = \{x | x \sim \mathcal{X}_{\mathcal{P}}\}$, on our standard MNIST and CIFAR10 experiments as the base cases. In details, we ran all eight binary combinations of using/not using $\mathcal{L}_{\mathrm{OH}}, \mathcal{L}_{\mathrm{IE}}, \mathcal{L}_{\mathrm{CT}}$ to optimize $\mathcal{D}$, and report in Fig. B.1. The contribution of each objective is derived by the mean difference in student accuracy between using/not using the corresponding loss, while standard error is aggregated

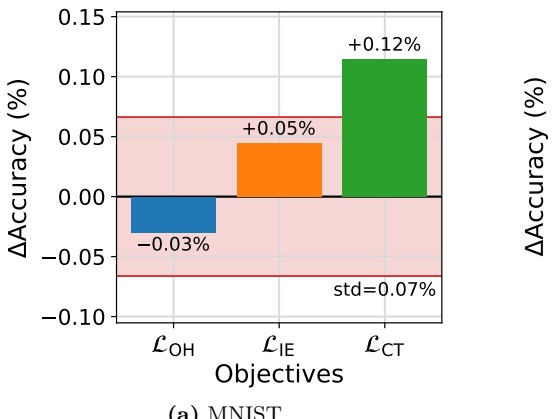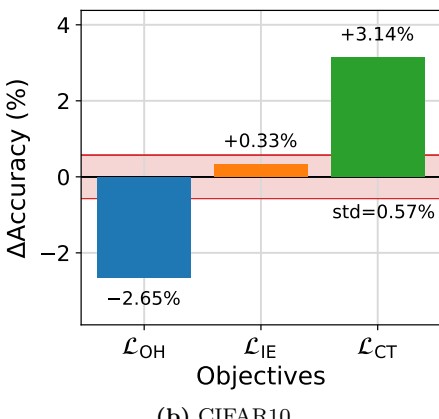

**(a)** MNIST

**(b)** CIFAR10

Figure B.1: Contribution to distillation accuracy (%) of one-hot loss ($\mathcal{L}_{\text{OH}}$), information entropy loss ($\mathcal{L}_{\text{IE}}$), and contrastive loss ($\mathcal{L}_{\text{CT}}$), on MNIST and CIFAR10.

on a per-combination basis. We observe $\mathcal{L}_{\text{IE}}$ only has modest contribution, as class-balance has already been achieved during Synthesis. Surprisedly, in-class similarity $\mathcal{L}_{\text{OH}}$ is actually adversarially destructive, harming KD performance by a significant amount of $-2.65\%$. Meanwhile, contrastive optimization using $\mathcal{L}_{\text{CT}}$ fruitfully helped gain KD performance, notably by $+0.12\%$ on MNIST and $+3.14\%$. Thus, we focus solely on diversity and employ only the contrastive loss $\mathcal{L}_{\text{CT}}$ to optimize image priors.

On a side note, Zhang et al. (2022); Yuan et al. (2024) employed $\mathcal{L}_{\text{OH}}$ and $\mathcal{L}_{\text{IE}}$ on the weights of a generator network, while we employ $\mathcal{L}_{\text{CT}}$ on the parameterized image priors $\mathcal{X}_{\mathcal{P}}$ and an instance-discriminator $R$. We speculate that this difference also affects the comparision and requires further investigation.

## C Extension to Iterative Framework

An interesting extension of our framework is performing *iterative Contrast-Distillation* to further boost student performance. Recall that in the Contrast phase (Sec. 3.3), we employ an primer student $S_0$ to optimize the dataset $\mathcal{D}$, and train a new student $S$ in the next Distillation phase. It naturally emerges that the stronger $S$ possesses even more robust embeddings and can be *recycled for Contrast phase again.* Iteratively, we may re-employ the latest student $S_g$ at the $g$-th generation in the role of a new primer student to optimize $\mathcal{D}$ in the next Contrast phase, and query soft labels from it to train the new student $S_{g+1}$ in the next Distillation phase. The objectives for $S_{g+1}$, extended from $\mathcal{L}_S$ (Eq. 10), is to match the teacher's label $T(x)$ with $\mathcal{L}_{\text{Hard-KD}}$, and to match previous student's logits $S_g^{\mathcal{Z}}(x)$ with $\mathcal{L}_{\text{Soft-KD}}$, on samples $x \sim \mathcal{D}$:

$$\mathcal{L}_{S_g} = \mathcal{L}_{\text{Hard-KD}} + \mathcal{L}_{\text{Soft-KD}} = \mathbb{E}_{x \sim \mathcal{D}} \left[ \mathcal{L}_{\text{CE}} \left( \hat{y}_{S_{g+1}-\text{Soft}}, \hat{y}_{T-\text{Hard}} \right) + \mathcal{L}_{\text{L1}} \left( S_{g+1}^{\mathcal{Z}}(x), S_g^{\mathcal{Z}}(x) \right) \right]. \quad \text{(C.1)}$$

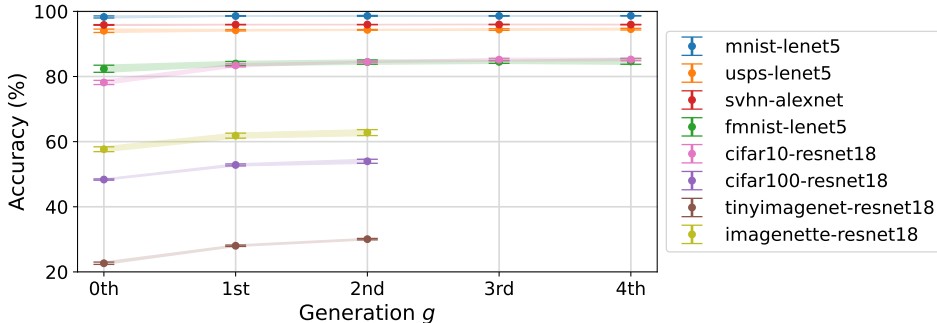

Figure C.1: Distillation accuracy (%) by generation $g$.

From our standard experiments (Sec. 4.3, corresponds to $g = 1$), we iterate for more generations to reach $g = 4$ on simple datasets and $g = 2$ on complex datasets due to resource constraints. As reported in Fig. C.1, even though KD performance converges quickly on simple dataset, it iteratively improves until the last generations with trivial fluctuations. These results conclude that our iterative Contrast-Distillation strategy can robustly deliver increasingly better students with more generations, and that generalization is not hurt with longer training. Nonetheless, computational cost increases *linearly* but student accuracy does not scale as fast to justify the trade-off. We look forward to making this extension even more efficient in future works.

# D Explicit Results

In this section, we provide explicit numerical results for the figures presented in the main text and above sections of the Appendix for completeness and reproducibility.

## D.1 Increasing synthetic dataset size

We provide explicit student accuracy and standard error for KD by number of samples, which has been visualized and discussed in Fig. 6, Sec. 4.4.4. The results are grouped by the simple datasets in Table D.1 and complex datasets in Table D.2.

Table D.1: Distillation accuracy (%) by number of sample $N$ on simple datasets.

| $N$ | MNIST LeNet5 | USPS LeNet5 | SVHN AlexNet | FMNIST LeNet5 |
|---|---|---|---|---|
| 10 K | $98.20_{\pm.06}$ | $93.22_{\pm.43}$ | $94.64_{\pm.13}$ | $76.14_{\pm2.48}$ |
| 20 K | $98.59_{\pm.08}$ | $93.99_{\pm.24}$ | $95.46_{\pm.07}$ | $79.49_{\pm1.38}$ |
| 30 K | $98.68_{\pm.03}$ | $94.05_{\pm.08}$ | $95.71_{\pm.05}$ | $82.67_{\pm.45}$ |
| 50 K | $98.75_{\pm.03}$ | $94.20_{\pm.21}$ | $95.94_{\pm.05}$ | $83.98_{\pm.65}$ |
| 70 K | $98.81_{\pm.03}$ | $94.25_{\pm.06}$ | $95.97_{\pm.02}$ | $84.84_{\pm.69}$ |
| 100 K | $98.78_{\pm.05}$ | $94.41_{\pm.08}$ | $96.06_{\pm.03}$ | $\mathbf{85.63}_{\pm.39}$ |
| 128 K | $\mathbf{98.90}_{\pm.02}$ | $\mathbf{94.44}_{\pm.09}$ | $\mathbf{96.13}_{\pm.01}$ | $85.62_{\pm.06}$ |

Best results are in **bold**.

Table D.2: Distillation accuracy (%) by number of samples $N$ on complex datasets.

| $N$ | CIFAR10 | CIFAR100 | TinyImageNet | Imagenette |
|---|---|---|---|---|
| 10 K | $63.54_{\pm2.64}$ | $25.85_{\pm1.31}$ | $07.88_{\pm.49}$ | $50.82_{\pm1.65}$ |
| 20 K | $72.98_{\pm.50}$ | $37.48_{\pm.48}$ | $12.13_{\pm.17}$ | $57.43_{\pm1.51}$ |
| 30 K | $78.18_{\pm.60}$ | $42.82_{\pm.56}$ | $13.96_{\pm.42}$ | $58.98_{\pm1.19}$ |
| 50 K | $83.41_{\pm.46}$ | $48.45_{\pm.26}$ | $16.49_{\pm.37}$ | $61.84_{\pm.77}$ |
| 70 K | $85.20_{\pm.35}$ | $50.75_{\pm.18}$ | $18.69_{\pm.63}$ | $63.08_{\pm.96}$ |
| 100 K | $86.60_{\pm.29}$ | $52.85_{\pm.30}$ | $19.99_{\pm.36}$ | $65.98_{\pm.78}$ |
| 128 K | $87.56_{\pm.11}$ | $53.69_{\pm.26}$ | $21.07_{\pm.50}$ | $\mathbf{66.60}_{\pm.72}$ |
| 256 K | $\mathbf{89.24}_{\pm.15}$ | $\mathbf{56.84}_{\pm.54}$ | $24.46_{\pm.26}$ | - |
| 512 K | - | - | $\mathbf{28.06}_{\pm.22}$ | - |

Best results are in **bold**.

## D.2 Students of compressed capacity

We provide in Table D.3 explicit student accuracy and standard error for KD by compression ratio, which has been visualized and discussed in Fig. 7, Sec. 4.4.7.

Table D.3: Distillation accuracy (%) by compression ratio (CR).

| CR | MNIST LeNet5 | USPS LeNet5 | SVHN AlexNet | FMNIST LeNet5 | CIFAR10 ResNet18 |
|---|---|---|---|---|---|
| 100% | $98.59_{\pm\ .08}$ | $94.20_{\pm\ .21}$ | $95.94_{\pm\ .05}$ | $83.98_{\pm\ .65}$ | $83.41_{\pm\ .46}$ |
| 25% | $96.79_{\pm1.39}$ | $92.42_{\pm\ .08}$ | $95.41_{\pm\ .04}$ | $74.90_{\pm\ .37}$ | $79.23_{\pm\ .51}$ |
| 4% | $85.83_{\pm1.86}$ | $90.70_{\pm\ .33}$ | $92.51_{\pm\ .07}$ | $52.55_{\pm3.79}$ | $64.48_{\pm\ .81}$ |
| 1% | $47.72_{\pm7.44}$ | $52.43_{\pm1.47}$ | $72.00_{\pm2.89}$ | $36.85_{\pm8.66}$ | $47.39_{\pm3.30}$ |

## D.3 Extension to Iterative Framework

We provide explicit student accuracy and standard error for KD by number of generations, which has been visualized and discussed in Fig. C.1, Sec. D.3. We ran the experiments for all datasets, but limits to $g = 2$ on complex dataset due to resource constraints, as shown in Table D.4. We additionally remark that the student can get iteratively better until the last generations, with some trivial fluctuations.

Table D.4: Distillation accuracy (%) by generation $g$.

| $g$ | MNIST LeNet5 | USPS LeNet5 | SVHN AlexNet | FMNIST LeNet5 | CIFAR10 ResNet18 | CIFAR100 ResNet18 | Tiny[*] ResNet18 | Nette[**] ResNet18 |
|---|---|---|---|---|---|---|---|---|
| 0 | $98.29_{\pm.32}$ | $94.04_{\pm.52}$ | $95.84_{\pm.10}$ | $82.38_{\pm1.09}$ | $78.17_{\pm.64}$ | $48.36_{\pm.19}$ | $22.66_{\pm.36}$ | $57.67_{\pm.74}$ |
| 1 | $98.59_{\pm.08}$ | $94.20_{\pm.21}$ | $95.94_{\pm.05}$ | $83.98_{\pm\ .65}$ | $83.41_{\pm.46}$ | $52.85_{\pm.30}$ | $28.03_{\pm.22}$ | $61.84_{\pm.77}$ |
| 2 | $98.62_{\pm.12}$ | $94.32_{\pm.12}$ | $95.95_{\pm.03}$ | $84.43_{\pm\ .66}$ | $84.50_{\pm.16}$ | $\mathbf{53.97}_{\pm.62}$ | $\mathbf{30.06}_{\pm.22}$ | $\mathbf{62.78}_{\pm.91}$ |
| 3 | $98.61_{\pm.08}$ | $94.42_{\pm.25}$ | $\mathbf{95.97}_{\pm.06}$ | $84.50_{\pm\ .46}$ | $85.18_{\pm.42}$ | - | - | - |
| 4 | $\mathbf{98.65}_{\pm.04}$ | $\mathbf{94.54}_{\pm.25}$ | $95.94_{\pm.06}$ | $\mathbf{84.62}_{\pm\ .86}$ | $\mathbf{85.23}_{\pm.39}$ | - | - | - |

Best results are in **bold**. [*]TinyImageNet, [**]Imagenette

