# OpenReview forum: "Diverse Image Priors for Black-box Data-free Knowledge Distillation"
_TMLR — Rejected by TMLR_

### Review · Reviewer_MB8i · 2025-11-13

**Summary Of Contributions:**

The paper explores the idea of diverse image priors for black-box knowledge distillation from teacher to student networks. The work proposes three step diverse synthetic image prior generation, followed by training primer student with contrastive objecting and finally distillation. The main contribution of the work is proposal of diverse image priors and training primer student for extracting soft labels of teacher network outputs, that was not used in black-box KD methods before.

**Audience:**

Yes

**Audience Explanation:**

I believe this paper presents several compelling claims about the impact of diverse image priors and the role of primer students in extracting soft features during the knowledge distillation process. In particular, given the challenging black-box setting, the evidence and insights provided appear both useful and impactful. The work also addresses a highly realistic scenario: researchers and engineers often have limited access to teacher models, yet the demand for effective distilled models continues to grow. By focusing on this practical constraint, the paper contributes to an increasingly relevant area of study.

**Broader Impact Concerns:**

Authors addressed ethical considerations in Section 5 of the manuscript.

**Claims And Evidence:**

No

**Claims Explanation:**

The paper presents an extensive set of experiments across both simple and challenging datasets. The selected baselines cover standard knowledge distillation methods as well as previously proposed black-box KD approaches. Overall, the results indicate that the proposed DIP-KD method outperforms the baselines in terms of distillation accuracy. The ablation study and compression ratio (CR) experiments are also informative and strengthen the empirical analysis.

---

However, **not all relevant baselines are included in the evaluation**. For instance, Table 1 omits BBKD baselines for ZSDB3 and IDEAL for USPS. While it is understandable that the authors source baselines directly from the respective papers, presenting the full set would more clearly illustrate the advantages of DIP-KD and ensure fair, transparent comparisons. A similar issue appears in Table 2. Although DIP-KD shows improvements of <1% over DFHL-RS on CIFAR-100 and 0.08% (which could be due to variance) over IDEAL on TinyImageNet, the more challenging Imagenette baseline is missing both of these methods. Without these entries, it becomes difficult to draw confident conclusions about whether the proposed method performs better, worse, or comparably to prior approaches. If the authors intend to claim improvements over all baselines, then the corresponding tables should include the complete set of comparisons to support these claims.

**Requested Changes:**

**Suggested Major Changes**
* **Provide complete baselines for Tables 1 and 2:** It is essential to include the full set of relevant baselines before making claims of superiority. Without complete comparisons, the conclusions remain incomplete and potentially misleading.

**Suggested Minor Changes**

* **Fix typos:** In contribution item no. 2, the phrase “making every images distinguishable” should likely read “making every set of images distinguishable”. Additionally, in Figure 1, the title contains a typo: “Synthetis” should be corrected to “Synthesis.”

* **Improve readability and formatting:** The paper can be difficult to follow due to the introduction of numerous definitions and variables. Streamlining expressions or dedicating a dedicated section for the most essential preliminaries would improve clarity. Furthermore, the placement of figures and tables can be confusing. For example, Table 5 appears directly under Figure 5, making it easy to overlook. Better spacing or repositioning would improve readability.

---

> ### Author Response · Authors · 2025-12-19
> **Official Comment of Authors to Reviewer MB8i**
>
> Thank you for the constructive and detailed feedback. We would like to address the suggested changes as below.
>
> ---
> ### Concern 1: Tables 1 and 2 do not include some baselines from other BBDFKD methods (ZSDB3 [1], IDEAL [2], DFHL-RS [3]).
>
> We agree with the reviewer that a comprehensive comparison is important for transparent and definitive evaluation. Thus, we have **reproduced baseline results directly from the published source code**. We also note that we run all methods under identical settings, and all baselines were tuned fairly using their official code and recommendations. We now feature full numerical results for all benchmarks in Tables 1 and 2 and will revise our discussion in the manuscript accordingly.
>
> ---
> ### Concern 2: Typos and minor corrections
> Thank you for catching these typos. We have corrected all noted issues:
> 1. Contribution item #2 now reads: *“We investigate contrastive learning as a fitting strategy to improve image diversity, where synthetic images are optimized to be distinguishable to each other.”*
> 2. Typo in Figure 1 text has been corrected from *“Synthetis”* to *“Synthesis”*.
>
> We have also proof-read the manuscript again to scan for any remaining typos.
>
> ---
> ### Concern 3: Readability and formatting. The paper is difficult to follow due to dense definitions, variables, and visual clutter.
> We acknowledge the concern regarding notation density. We have made the following improvements in the revised version:
> 1. Condensing the preliminaries to an individual, dedicated “Problem Setup” subsection
> 2. Simplifying several expressions in Sections 3.2 and 3.3 for clarity while preserving the rigor of the original formulation.
> 3. Repositioning figures/tables to avoid visual clutter. For example, Table 5 and Figure 5 are no longer placed directly adjacent and are discussed separately.
>
> ---
> We kindly thank the reviewer again for the feedbacks.
>
> References:
> 1. Zero-shot Knowledge Distillation from a Decision-based Black-box Model - Wang, ICML 2021.
> 2. IDEAL: Query-Efficient Data-Free Learning from Black-Box Models - Zhang et al., ICLR 2023.
> 3. Data-free hard-label robustness stealing attack - Yuan et al., AAAI 2024.

---

> > ### Comment · Reviewer_MB8i · 2025-12-23
> >
> > I thank the authors for providing clearer baselines, which better highlight the aspects in which the proposed method outperforms previous implementations. I would like to follow up on another comment made by Reviewer 95y4 regarding hyperparameter tuning.
> >
> > I see that some parameters are chosen automatically, but others ($\beta$, $M$) are still selected manually. The authors claim the following in their rebuttal:
> >
> > > Each pass forms clearer "blobs", so that we obtain cohesive shapes in $\hat{m}$. We use fixed $M = 10$ for a balance between diversity and complexity
> >
> > I think these explanations are very loose and require further elaboration. What exactly is the complexity tradeoff here? To what extent does the method become slower when using $M > 10$? Moreover, the ablation study (Table 9) suggests that $M > 10$ is not necessarily better and varies significantly depending on the dataset. So how exactly should this hyperparameter be chosen? Are there any guidelines or rules of thumb?
> >
> > > $\beta = 0.5$ naturally yields balanced positive/negative regions in the masks $\hat{m}$ (Fig. 2), allowing our method to be generalizable and interpretable
> >
> > Again, I would like the authors to expand further on the parameter $\beta$. In which scenarios does lower/higher threshold between positive and negative regions of the mask yield better performance? The authors' response alludes to the fact that the parameters are tuned to the data, which circles back to Reviewer 95y4’s concern about violating the data-free assumption.

---

> ### Author Response · Authors · 2025-12-23
> **Official Comment by Authors to Reviewer MB8i**
>
> We thank the reviewer for the follow-up and clarify that the hyperparameters $\beta$ and $M$ **do not require tuning** and **were not tuned on any dataset**.
>
> ---
> ### Choice of $M$: a robust constant
> In our newly revised Section 4.4.8, we add an ablation study on $M$ and clarify that $M$ controls the ***geometric complexity*** of mask $\hat{m}$, not its *computational complexity* (cost scales linearly and is negligible). Particularly, Fig. 8 illustrates the effect of $M$ on $\hat{m}$:
> + For low $M$ < 5, mask $\hat{m} \approx m_{0}$ and exhibits salt-and-pepper patterns that are too fine-grained to form cohesive regions. This leads to pixel-wise noisy cutmix results, which is *semantically not diverse*.
> + At high $M$ > 20, $\hat{m}$ collapses into overly simple blobs, yielding highly similar masks and again *reducing diversity* after cutmixing.
>
> A broad range $M\in[5,20]$ produces stable and diverse mask structures. We fix $M$ = 10 across **all benchmarks** (Tables 1 and 2), demonstrating that a single constant works robustly without dataset-specific tuning. For completeness, we note that higher $M$ can be more suitable for simple domains (e.g., $M$ = 20 on MNIST in Table 9), whereas lower $M$ may be useful for noisy, complex domains. However, this does not contradict our claim: $M$ = 10 is a **strong out-of-the-box choice** and does not require optimization.
>
> ---
> ### Choice of $\beta$: a structural design
> The parameter $\beta$ = 0.5 is **structural and not a free hyperparameter**. Deviations from this value effectively **negates the cutmix step**, reducing the method to the remaining components. It follows directly from the formulation of the diverging filter $f_{\text{divg}}$​ (Appendix A.3.1). When $\beta \approx 0$ or $\beta \approx 1$, each pass through $f_{\text{divg}}$ monotonically brightens or darkens the mask. Because mask formation requires multiple passes, any skewed $\beta \ne 0.5$ compounds this effect and empirically collapses $\hat{m}$ towards being near-white or near-black. Cutmixing with these masks is mostly equivalent to return one of the two input images, which disables the cutmix step itself.
>
> ---
> In summary, $M$ is a robust default constant that generalizes well without tuning, and $\beta$ is structurally determined. They do not require dataset-specific tuning, nor violate the data-free assumption. We will incorporate these clarifications into the final revision.

---

### Review · Reviewer_95y4 · 2025-11-24

**Summary Of Contributions:**

The paper studies the problem of Data-Free Knowledge Distillation. The authors propose a synthetic data generation method that utilizes a "diverging filter" and hierarchical noise to encourage diversity in the generated samples. The proposed method is extensively evaluated on standard image classification benchmarks using white-box teacher architectures.

**Additional Comments:**

N/A

**Audience:**

No

**Audience Explanation:**

The paper's generalization capability and practical applicability are limited by disconnects between its motivation, methodology, and evaluation.

   First, the experimental validation relies exclusively on white-box teachers despite the motivation being grounded in black-box scenarios (e.g., LLMs), creating a fundamental gap between the problem statement and the proposed solution.

   Second, the methodology's focus on maximizing diversity via "diverging filters" is ill-suited for the specific, narrow-manifold domains cited in the introduction (such as medical imaging), where it risks generating out-of-distribution samples that fail to capture class-conditional densities.

   Finally, the method's reliance on sensitive hyperparameters (e.g., $\beta$) undermines its robustness in true "data-free" settings, as there is no clear mechanism to tune these parameters without access to a validation set, thereby violating the core assumption of the task.

**Claims And Evidence:**

No

**Claims Explanation:**

1. One of the motivations for developing black-box data-free KD as the authors mentioned in the Introduction is that many teacher models (e.g., GPT, Claude) are black-box models. However, all experiments in this paper use white-box teachers (LeNet, AlexNet, ResNet) with full access to internal representations. This evaluation mismatch undermines the practical relevance of the work.

2. The method relies on specific hyperparameters, such as the thresholds for the "diverging filter" (Eq. 5) and the specific scales for hierarchical noise. In a real-world data-free setting, how are these hyperparameters tuned? Note that the $\beta$ and the number of the CutMix iteration are fixed in this work. As Fig.A.1 shows, $\beta$ is not an insignificant parameters. This setting absolutely can not generalize well to all scenarios. Usually, one needs a validation set to tune $\beta$ or the number of CutMix iterations. If they tuned these on the `target datasets` to get the results in the tables, it violates the "data-free" assumption. The paper does not explicitly explain how hyperparameters can be selected without access to validation data.

3. I think there exists a misalignment between the motivation and methodology. The authors motivate their approach by citing privacy-sensitive applications such as medical imaging (health data) and user profiles (personal data). However, these domains are characterized by `highly specific, narrow distributions` rather than diverse, general-purpose distributions (like ImageNet). For instance, medical images reside on a tight, low-dimensional manifold with strict structural constraints, where "diversity" is often noise rather than semantic variation. Merely "diverging" the input space without respecting the underlying class-conditional density is likely to generate Out-of-Distribution (OOD) synthetic samples that fall off the real data manifold. The claim that this method is suitable for the motivated scenarios (health/personal data) is unsupported.

**Requested Changes:**

1. Define the `diversity` in the abstract and introduction sections before or after you mention it.
2. The current Abstract is loosely structured. The coherence of the abstract needs to be improved. For example, there is a lack of connection between the sentences. These challenges have inspired black-box data-free KD, in which only the teacher’s top-1 predictions and no real data are available. While recent approaches tend to synthetic data, they largely overlook data diversity, which is crucial for effective knowledge transfer.
3. Can you apply your DIP-KD method to black-box teacher models and report the performance? This would significantly strengthen the paper's claims and practical relevance.

---

> ### Author Response · Authors · 2025-12-19
> **Official Comment of Authors to Reviewer 95y4**
>
> We sincerely thank the reviewer for the thoughtful comments and the opportunity to clarify several aspects of our work.
>
> ---
> ### Concern 1: Experiments use LeNet/AlexNet/ResNet, which are "white-box", contradicting the black-box motivation.
> We thank the reviewer for raising this important point, and we appreciate the chance to clarify our **strict adherence to the black-box setting**. The misunderstanding may stem from terminology: in KD literature, the white-/black-box distinction is determined by the accessible interface, not by the network’s architecture. In our setup, the teacher provides only **top-1 hard labels**, and **no internal information is ever exposed** (including features, logits, or gradients). This protocol follows established black-box KD practice in ZSDB3 [1], IDEAL [2], DFHL-RS [3]). Our black-box implementation can be verified at https://osf.io/5mry8/?view_only=dee9e8fbcd114c34b45aa958a3aa32fa.
>
> ---
> ### Concern 2: The method appears to rely on tunable hyperparameters, raising concerns about validation data.
> Thank you for pointing this out. We respectfully clarify that our method does **not** rely on tuning on any datasets. The parameters in question are either **determined automatically** from image dimension, **structural design choice**, or **robust constants**. We explain how they behave in our pipeline:
>
> 1. Image scales in **Hierarchical Noise** (Section 3.1.1): The scale set is derived automatically from the image dimension. In details, we sample powers-of-two scales until they fully cover the intended dimension (Eq. 2). For instance, synthesizing a $32 \times 32$ image yield scales $\\{2^0, 2^1, \ldots, 2^5\\}$.
> 2. Semantic mask hyperparameters in **Semantic CutMixing** (Section 3.1.3.): By motivation, Semantic Cutmixing is effective because it mixes images to create **new hybrids with diverse semantics**, not fine-grained parameter choice. We clarify that the parameters in question were not tuned on any datasets, but are chosen for robust formation of semantic masks $\hat{m}$:
> 	+ $\beta$ **= 0.5**: We use $\beta$ as a **fixed structural parameter** to control the threshold of positive/negative pixels. Setting $\beta$ = 0.5 naturally yields balanced positive/negative regions in the masks $\hat{m}$ (Fig. 2), allowing our method to be generalizable and interpretable.
> 	+ **Number of filter iterations = 10**: This parameter, hereby denoted as $M$, is the number of steps the we put a random mask $m_{0}$ through the filters. Each pass forms clearer "blobs", so that we obtain cohesive shapes in $\hat{m}$. We use fixed $M$ = 10 for a balance between diversity and complexity, allowing generalizability. To highlight this more clearly, we have added a dedicated ablation study on $M$, giving robust mask formation and KD performance for $M \\in [5, 20]$.
>
> ---
> ### Concern 3: Synthetic samples may fall out-of-distribution (OOD) of real data in specific domains
> We sincerely thank the reviewer for pointing out this concern. While we agree that certain domains of narrow distributions are difficult to replicate, the ultimate goal of KD is **student accuracy** where data serve as a medium. What matters most is that our synthetic data can **cover the real data distribution** for effective KD. We acknowledge OOD data only causes inefficiency issues, as what is learned on OOD data is irrelevant during evaluation.
>
> We back this claim by showing that our synthetic data is a **mix of in-distribution and OOD data**. In Section 4.4.2, we visualize data distribution in Fig. 3 and measure distribution metrics in Table 4, showing that our synthetic samples has good coverage of the real counterparts. Thus, they can serve as a robust medium to perform KD. We believe diversity is a well-motivated for data-free scenarios, even in narrow domains such as privacy-sensitive applications.
>
> ---
> ### Concern 4: Lack of diversity definition
> We thank the reviewer for this insight. We have added the definition for diversity in the Abstract and Introduction, and this helps the formulation to be more coherent. We also note that while diversity is mentioned in previous works (IDEAL [2], DFHL-RS [3]), yet was typically not adequately evaluated. In contrast, we provide both qualitative and quantitative evaluations in Section 4.4.2 and 4.4.3.
>
> ---
> ### Concern 5: The Abstract is loosely structured.
> Thank you for this comment. We appreciate the suggestion and have refined the abstract wording for clarity.
>
> ---
> We sincerely thank the reviewer again for the feedbacks and will reflect these changes accordingly in our revision. We hope that our work provides clear evidence and may be useful to researchers in future knowledge distillation studies.
>
> References:
> 1. Zero-shot Knowledge Distillation from a Decision-based Black-box Model - Wang, ICML 2021.
> 2. IDEAL: Query-Efficient Data-Free Learning from Black-Box Models - Zhang et al., ICLR 2023.
> 3. Data-free hard-label robustness stealing attack - Yuan et al., AAAI 2024.

---

> > ### Comment · Reviewer_95y4 · 2025-12-23
> > **Thanks for your reply, but some concerns remain**
> >
> > Thanks for addressing some of my concerns. However, I think the following points remain unaddressed:
> >
> > 1. Regarding the black-box teacher concern, I raised this point because the motivation in the introduction emphasized black-box scenarios (e.g., several large-scale vision models are highlighted in the `4-th paragraph of the Introduction`). However, all experiments used white-box teachers with full access to internal representations. This creates a disconnect between the problem statement and the proposed solution.
> > 2. Regarding the hyperparameter tuning concern, I think the Reviewer `MB8i` has also noticed this point and made a clear question about it.
> > 3. Regarding the motivation concern, your argument relies on distribution metrics (Table 4) and embedding visualizations (Fig. 3) from *general-purpose public datasets* (e.g., CIFAR10, TinyImageNet). These datasets are explicitly designed to **capture broad, diverse semantic variations** (e.g., animals, vehicles, everyday objects) **with relatively loose structural constraints**. For such data, your "diverse image priors" (via hierarchical noise, nonlinear transforms, and semantic cutmixing) can naturally overlap with the real data manifold **because the manifold itself is wide and flexible**. But for the dataset that are in such diverse domains, we don't need data-free KD methods, since we can easily collect data from the internet.
> >
> >    However, privacy-sensitive domains like medical imaging or personal user data **as you mentioned in the motiviation** of the `3-rd paragraph of the Introduction` section have **radically narrower, more structured manifolds**.
> >
> >    Your current experiments provide no evidence that your method actually works in these scenarios or can avoid generating OOD samples in such narrow domains. The metrics you report (i.e., the diverse image priors you designed) only seem to **reflect general distributional coverage on already-diverse datasets**.
> >
> >    The "coverage" demonstrated on CIFAR-10/TinyImageNet simply does not translate to domains with tight domain-specific manifolds, which are precisely what you claim to focus on in the introduction. So I'm not convinced that the motivation you lay out upfront is actually supported by your methodology and experiments.

---

> ### Author Response · Authors · 2025-12-25
> **Official Comment by Authors to Reviewer 95y4**
>
> Thank you for the thoughtful follow-up. We appreciate the opportunity to clarify these points and address the remaining concerns.
>
> ---
> ### 1. The black-box teacher setting
> We clarify that our method is **strictly black-box**. Specifically, we describe it in our *Problem Setup* (Sec. 3.1) as: "*Given solely a **black-box** pre-trained teacher network $T$ that returns only top-1 hard labels (no internal features, logits, or gradients) [...]*". While the teacher architectures are known for reproducibility, **no internal signals are ever accessed during distillation**.
>
> We kindly suggest that our black-box setup can be verified in two ways:
> 1. Via previous literatures: Our setup aligns with black-box data-free KD methods ZSDB3 [1], IDEAL [2], and DFHL-RS [3].
> 2. Via source code: Our released code similarly reflects this practice and is published at https://osf.io/5mry8/?view_only=dee9e8fbcd114c34b45aa958a3aa32fa.
>
> ---
> ### 2. Hyperparameter tuning
> We emphasize that neither $\beta$ nor $M$ was tuned, as these parameters do not require dataset-specific adjustment. As discussed in our response to Reviewer MB8i, $\beta$ is structurally fixed by design, and $M$ is used as a single robust constant across all benchmarks.
>
> ---
> ### 3. Applicability on privacy-sensitive domains
> We agree that general-purpose datasets exhibit broad semantic variability and are not proxies for narrowly structured or privacy-sensitive domains. However, our primary contribution is **Black-box Data-free KD**, independent of any particular domain. We adopt general-purpose datasets as **standard evaluation benchmarks in black-box data-free KD** for direct comparison with prior works, rather than as proxies for specific domains.
>
> We further note how **privacy-sensitive domains can be diverse** as well (in our revised *Introduction*), such that our diversity-driven approach remains useful in such settings. To directly address this concern, we additionally perform KD on multiple modalities of the medical dataset **MedMNIST** [4] (Table R1). We compare NaiveKD, DFHL-RS (the strongest baseline), and our DIPKD; where **DIPKD consistently achieves the highest accuracy with clear margins** across all datasets.
>
> **Table R1**: Distillation accuracy (%) on medical datasets of MedMNIST
>
> | | BreastMNIST | OrganAMNIST | DermaMNIST | BloodMNIST |
> | --- | --- | --- | --- | --- |
> | Teacher | 79.49 | 87.60 | 79.60 | 98.19 |
> | NaiveKD | 61.78$_{\pm 1.60}$ | 24.09$_{\pm 0.44}$ | 56.82$_{\pm 1.50}$ | 9.09$_{\pm 0.13}$  |
> | DFHL-RS | 73.81$_{\pm 1.80}$ | 75.12$_{\pm 2.89}$ | 64.71$_{\pm 3.02}$ | 61.67$_{\pm 3.11}$ |
> | DIPKD   | **77.95**$_{\pm 0.77}$ | **82.32**$_{\pm 1.08}$ | **71.29**$_{\pm 2.67}$ | **73.98**$_{\pm 1.77}$ |
>
> (Subscript $_{\pm\ldots}$ denotes the standard error; Best results are in **bold**.)
>
> Importantly, the **primary objective in KD is accuracy**. Our method does not aim to approximate or recover any domain-specific data manifold, which is infeasible by definition in the data-free setting. From this perspective, explicitly curating samples toward any narrow domain would require prior domain knowledge and therefore **contradicts the black-box data-free assumption**.
>
> ---
>
> We thank the reviewer for the constructive feedback and will incorporate these clarifications and additional results in the final revision.
>
> References (continued):
> + 4. MedMNIST Classification Decathlon: A Lightweight AutoML Benchmark for Medical Image Analysis - Yang et al., ISBI 2021

---

### Review · Reviewer_a72P · 2025-12-08

**Summary Of Contributions:**

The paper tackles the problem of black-box data-free knowledge distillation (BBDFKD). In this problem, the teacher gives access to the top-1 labels, and there are no labels available. To tackle this problem, the authors propose a 3-phase contribution to: 1. Synthesise image priors, 2. train a primer student ($S_0$) on the synthetic data using hard labels from the teacher, 3. train the final student ($S$) on the optimised synthetic data. The paper is evaluated among 8 popular datasets. The main idea of the paper is the way synthetic data is constructed and optimised through a 3-stage process. Overall, the paper tackles a clear problem setup using a simple pipeline with strong empirical performance. However, I have some concerns about different parts of the paper, which I detail in the next sections.

**Additional Comments:**

N/A.

**Audience:**

Yes

**Audience Explanation:**

The topic of the paper, black-box, data-free knowledge distillation, can have a broader impact on the community. Given that many of the current SOTA models tend to have accessibility restrictions, such as proprietary LLMS. While the paper is somewhat specialised and focused on image classification and KD, these themes are broad enough that a non-trivial subset of TMLR’s readership. The findings of the paper, such as the role of diverse synthetic image priors in improving KD performance without data or full model access, offer modest but practical insights into transfer learning principles, which could appeal to researchers working on resource-constrained deployment, privacy-preserving ML, or generative methods for distillation.

**Broader Impact Concerns:**

The paper includes a dedicated "Ethical considerations" section (Section 5) that sufficiently addresses the primary broader impact concerns associated with this line of research. While the statement is robust regarding privacy and security, it does not explicitly discuss Bias Propagation. In a black-box data-free setting, the student model blindly mimics the teacher's decision boundaries. If the teacher model contains algorithmic biases (e.g., racial or gender bias), the student will propagate and potentially amplify these biases.

**Claims And Evidence:**

No

**Claims Explanation:**

The claims mentioned in the paper are supported through extensive experiments. Specifically, the claim that the proposed DIP-KD synthesises semantically diverse image priors and achieves SOTA performance is evidenced by Figure 2 and Table 3. The claim that existing methods lack diversity and DIP-KD improves it is supported in Table 4. Also, Figures 2 and 3 (UMAP) show qualitative analysis of the claim that image priors are diverse in hierarchical structures, nonlinearity, and semantics.

**Requested Changes:**

1. First, the initial premier student $S_0$ is trained on the initial synthetic data using the hard labels from the teacher. How do you ensure these generated data do not have a poor representation, especially given that the contrastive learning objective is also trained on this data? Overall, how do you ensure the accuracy of the primer student before the Contrast phase?

2. The paper assumes there is no access to the teacher model. If so, how are the diversity analyses (Figures 3–5, Tables 4–5) generated? Since I assume these are using the embeddings of the teacher model.

3. The numbers for the accuracies of the baselines are directly taken from their paper, not re-implemented in the same codebase. How do you ensure that the comparisons are fair, given that this may obscure differences in the training setup? I would suggest that the authors re-implement at least one strong baseline (e.g., IDEAL or DFHL-RS) in their codebase with shared teacher/student architectures and synthetic budget, and report the results side-by-side.

4. Please make sure the submission does not include typos, such as: "abd employ" (should be "and employ" - Page 21).

5. Given the focus on synthetic data and label-only black-box extraction, I would expect at least some discussion and, ideally, empirical comparison to more recent DFKD and extraction/MI methods such as NAYER (Tran et al., CVPR 2024) [1], and label-only model inversion methods like LOKT (Nguyen et al., NeurIPS 2023) [2].

[1] Tran, Minh-Tuan, et al. "Nayer: Noisy layer data generation for efficient and effective data-free knowledge distillation." Proceedings of the IEEE/CVF conference on computer vision and pattern recognition. 2024.

[2] Nguyen, Bao-Ngoc, et al. "Label-only model inversion attacks via knowledge transfer." Advances in neural information processing systems 36 (2023): 68895-68907.

---

> ### Author Response · Authors · 2025-12-19
> **Official Comment of Authors to Reviewer a72P**
>
> We sincerely appreciate the reviewer’s constructive feedbacks and the chance to address the raised concerns.
>
> ---
> ### Concern 1: The quality of the initial synthetic data and the primer student.
>
> We agree that the initial synthetic data (from image priors $\mathcal{X}_\mathcal{P}$) and the primer student $S_0$ plays a vital role in the downstream Contrast and Distillation phases. We clarify how we achieve this in our framework:
> 1. Our synthetic images are **not arbitrary or unstructured noise** but are designed with diversity-inducing components in our Synthesis pipeline, so that they are diverse in both visuals and semantics. We demonstrate this in Section 4.4.2: $\mathcal{X}_\mathcal{P}$ have high coverage of the distribution of held-out real datasets.
> 2. We expect that when $\mathcal{X}_\mathcal{P}$ is diverse, the primer student $S_0$ can achieve **decent performance** as well. In our per-component ablation study (Table 3), the *Synthesis* component has boosted $S_0$ to accuracy 98% on MNIST and 77% on CIFAR10, approximating the second-best baseline. This justifies that $S_0$ is useful for downstream phases Contrast and Distillation.
> 3. In the Contrast phase, we posit that even if $S_0$ might not have top-notch performance, it is still **fit for contrastive learning**. This is supported by the setup in popular contrastive learning frameworks [1-3]: which train from scratch and do not depend on a high-accuracy network.
>
> ---
> ### Concern 2: Diversity analyses seemingly relying on teacher model embeddings.
>
> We apologize for the unclear setup. We confirm that we employed the teacher's backbone in these ablation studies, but **only for evaluation purpose**. As we stated in Sections 4.4.2 and 4.4.3: we extract embeddings *"[..] from the backbone of the ResNet18 teacher network pre-trained on CIFAR10"*. This is indeed the ResNet18 teacher of 95.21% accuracy on CIFAR10, being reused as a held-out "oracle". We will explicitly highlight this details in our revision.
>
> ---
> ### Concern 3: Fairness of baseline comparisons when using reported numbers from original papers.
> We appreciate the reviewer highlighting this important point. We agree that reproducing baselines where possible improves transparency and comparability. In response, we have **reproduced the results with the published source code** of all three baseline methods: ZSDB3 [4], IDEAL [5], DFHL-RS [6]. To guarantee a fair comparison, we use the exact same teacher/student architectures, synthetic data budget as our method DIP-KD. We ensured that all baseline implementations were tuned ethically and fairly, following the authors’ recommendations and publicly released code. The results align closely with the original works. We now feature full numerical results in Tables 1 and 2, and will explicitly provide the link to baseline codes.
>
> ---
> ### Concern 4: Typos and writing issues.
>
> We thank the reviewer for pointing out the typo. We have thoroughly proofread the manuscript and corrected the minor grammatical issues.
>
> ---
> ### Concern 5: Missing discussion and comparison of relevant methods (NAYER and LOKT).
>
> We appreciate this suggestion and agree that these recent methods offer valuable context. Yet, **our problem setting slightly differs**: we focus on **black-box data-free KD**, while NAYER [7] works in a **gray-box** setting (it still requires the teacher's gradients) and LOKT [8] requires a **public real dataset** (we restrict to *no real data*). Thus, these two methods are not directly comparable in our evaluation setup. However, we appreciate that these references paint a more complete view to our problem, and will add them to our literature review.
>
> ---
> ### Concern 6: On bias propagation in black-box KD as an ethical consideration
>
> We appreciate the reviewer’s insightful suggestion. Indeed, in a purely label-only setting, the student inherits the teacher’s decision boundaries and may inadvertently amplify biases. We have added an ethical discussion (Section 5) to discuss bias propagation, its societal impact, and mitigations.
>
> ---
> We hope these answers and newly added results address the reviewer’s concerns. We thank the reviewer again for the constructive and encouraging feedback.
>
> References:
> 1. A Simple Framework for Contrastive Learning of Visual Representations - Chen et al., ICML 2020.
> 2. Momentum Contrast for Unsupervised Visual Representation Learning - He et al., CVPR 2020.
> 3. Barlow Twins: Self-Supervised Learning via Redundancy Reduction - Zbontar et al., ICML 2021.
> 4. Zero-shot Knowledge Distillation from a Decision-based Black-box Model - Wang, ICML 2021.
> 5. IDEAL: Query-Efficient Data-Free Learning from Black-Box Models - Zhang et al., ICLR 2023.
> 6. Data-free hard-label robustness stealing attack - Yuan et al., AAAI 2024.
> 7. NAYER: Noisy Layer Data Generation for Efficient and Effective Data-free Knowledge Distillation - Tran et al., CVPR 2024.
> 8. Label-Only Model Inversion Attacks via Knowledge Transfer - Nguyen et al., NeurIPS 2023.

---

### Author Response · Authors · 2025-12-19
**Official Comment by Authors**

We warmly thank all reviewers for their thoughtful and constructive feedback. Based on your suggestions, we have updated our revision to address the points raised. We are happy to provide any additional clarification or answer further questions you may have.

---

### Decision · Action_Editor_mRxo · 2026-02-01

**Recommendation:** Reject

**Audience:**

Yes

**Audience Explanation:**

The paper addresses black-box data-free knowledge distillation, a topic relevant to parts of the TMLR community, and proposes a synthetic data–based approach that would be of interest to researchers working on data-free learning and model distillation.

**Claims And Evidence:**

No

**Claims Explanation:**

The submission addresses a relevant problem in black-box data-free knowledge distillation and proposes a technically thoughtful synthesis and distillation pipeline. The empirical results on standard and general image datasets are competitive and suggest that the approach has merit. However, as stated in the reviewers’ final assessments, the evidence presented does not provide fully convincing support for some main claims made in the paper’s introduction.

Specifically, the reviewers noted that even after the revision, two concerns remain. First, the introduction emphasizes privacy-sensitive and medical applications as motivating scenarios, but the empirical validation and accompanying analyses are largely conducted on general-purpose public datasets. As a result, it is not clearly established that the proposed “diverse image priors” are appropriate or effective for narrow, structured domains of the type highlighted in the paper's motivation. Second, reviewers raised concerns about the clarity and robustness of key synthesis parameters. Although the authors clarified that these parameters are fixed in the reported experiments, the manuscript states that the number of mask-formation iterations is selected based on visual inspection and acknowledges that optimal values may vary across datasets, with tuning recommended when domain access is available. Given the remaining concerns and reviewers' final recommendation, these issues make it difficult to conclude that the claims are completely supported by convincing and clearly established evidence at this stage. Addressing these concerns in a future submission, for example, by more closely aligning the motivating applications with the empirical evaluation and by providing sufficiently detailed analysis of the parameter selection, would help strengthen the support for the paper’s main claims.

**Resubmission Of Major Revision:**

The authors may consider submitting a major revision at a later time.